# PHYRECON: Physically Plausible Neural Scene Reconstruction

**Junfeng Ni**[1,2][⋆][†], **Yixin Chen**[2][⋆], **Bohan Jing**[2], **Nan Jiang**[2,3], **Bin Wang**[2],
**Bo Dai**[2], **Puhao Li**[1,2], **Yixin Zhu**[3], **Song-Chun Zhu**[1,2,3], **Siyuan Huang**[2]
[⋆] Equal contribution [†] Work done as an intern at BIGAI
[1] Tsinghua University [2] State Key Laboratory of General Artificial Intelligence, BIGAI
[3] Peking University
https://phyrecon.github.io

## Abstract

We address the issue of physical implausibility in multi-view neural reconstruction. While implicit representations have gained popularity in multi-view 3D reconstruction, previous work struggles to yield physically plausible results, limiting their utility in domains requiring rigorous physical accuracy. This lack of plausibility stems from the absence of physics modeling in existing methods and their inability to recover intricate geometrical structures. In this paper, we introduce PHYRECON, the first approach to leverage both differentiable rendering and differentiable physics simulation to learn implicit surface representations. PHYRECON features a novel differentiable particle-based physical simulator built on neural implicit representations. Central to this design is an efficient transformation between SDF-based implicit representations and explicit surface points via our proposed Surface Points Marching Cubes (SP-MC), enabling differentiable learning with both rendering and physical losses. Additionally, PHYRECON models both rendering and physical uncertainty to identify and compensate for inconsistent and inaccurate monocular geometric priors. The physical uncertainty further facilitates physics-guided pixel sampling to enhance the learning of slender structures. By integrating these techniques, our model supports differentiable joint modeling of appearance, geometry, and physics. Extensive experiments demonstrate that PHYRECON significantly improves the reconstruction quality. Our results also exhibit superior physical stability in physical simulators, with at least a 40% improvement across all datasets, paving the way for future physics-based applications.

## 1   Introduction

3D scene reconstruction is fundamental in computer vision, with applications spanning graphics, robotics, and more. Building on neural implicit representations [45], previous methods [18, 63, 77] have utilized multi-view images and monocular cues to recover fine-grained 3D geometry via volume rendering [10]. However, these approaches have overlooked physical plausibility, limiting their applicability in physics-intensive tasks such as embodied AI [17, 1, 30] and robotics [15, 26, 60].

The inability to achieve physically plausible reconstruction arises primarily from the *lack of physics modeling*. Existing methods based on neural implicit representation rely solely on rendering supervision, lacking explicit incorporation of physical constraints, such as those from physical simulators. This oversight compromises their ability to optimize 3D shapes for stability, as the models are not informed by the physics that would ensure realistic and stable structures in the real world.

Additionally, these methods often *ignore thin structures*, focusing instead on optimizing the substantial parts of objects within images. This limitation is due to overly-smoothed and averaged optimization

38th Conference on Neural Information Processing Systems (NeurIPS 2024).

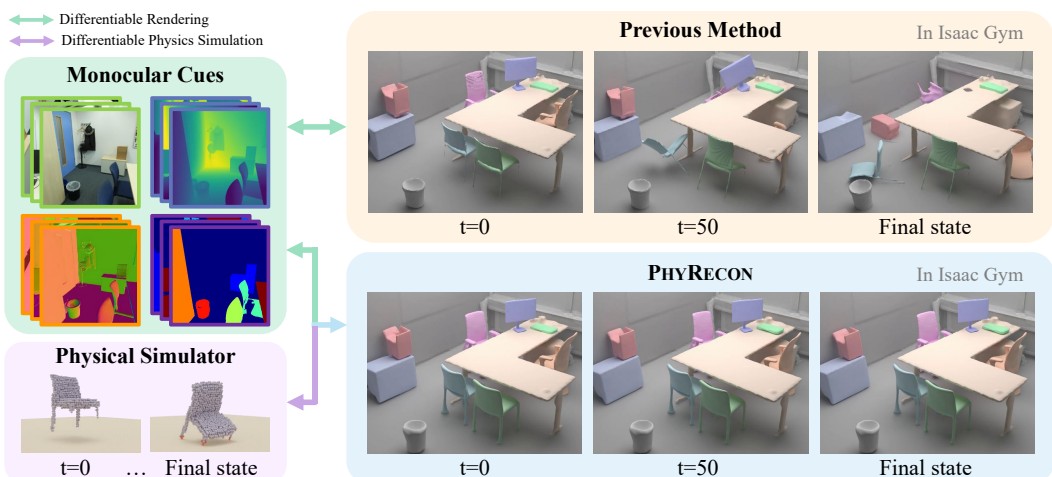

Figure 1: **Illustration of PHYRECON.** We leverage both differentiable physics simulation and differentiable rendering to learn implicit surface representation. Results from previous methods [32] fail to remain stable in physical simulators or recover intricate geometries, while PHYRECON achieves significant improvements in both reconstruction quality and physical plausibility.

results from problematic geometric priors, including multi-modal inconsistency (*e.g.*, depth-normal), multi-view inconsistency, and inaccurate predictions deviating from the true underlying 3D geometry. As a result, they struggle to capture the slender structures crucial for object stability, leading to reconstructions that lack the fine details necessary for accurate physical simulation.

To address these limitations, we present PHYRECON, the first effort to integrate differentiable rendering and differentiable physical simulations for learning implicit surface representations. Specifically, we propose a differentiable particle-based physical simulator and an efficient algorithm, Surface Points Marching Cubes (SP-MC), for differentiable transformation between SDF-based implicit representations and explicit surface points. The simulator facilitates accurate computation of 3D rigid body dynamics subjected to forces of gravity, contact, and friction, providing intricate details about the current shapes of the objects. Our differentiable pipeline efficiently implements and optimizes the implicit surface representation by coherently integrating feedback from rendering and physical losses.

Moreover, to enhance the reconstruction of intricate structures and address geometric prior inconsistencies, we propose a joint uncertainty model describing both rendering and physical uncertainty. The rendering uncertainty identifies and mitigates inconsistencies arising from multi-view geometric priors, while the physical uncertainty reflects the dynamic trajectory of 3D contact points in the physical simulator, offering precise and interpretable monitoring of regions lacking physical support. Utilizing these uncertainties, we adaptively adjust the per-pixel depth, normal, and instance mask losses to avoid erroneous priors in surface reconstruction. Observing that intricate geometries occupying fewer pixels are less likely to be sampled, we propose a physics-guided pixel sampling based on physical uncertainty to help recover slender structures.

We conduct extensive experiments on real datasets including ScanNet [9] and ScanNet++ [76], and the synthetic dataset Replica [62]. Results demonstrate that our method significantly surpasses all state-of-the-art methods in both reconstruction quality and physical plausibility. Through ablative studies, we highlight the effectiveness of physical loss and uncertainty modeling. Remarkably, our approach achieves significant advancements in stability, registering at least a 40% improvement across all examined datasets, as assessed using the physical simulator Isaac Gym [39].

In summary, our main contributions are three-fold:

1. We introduce the first method that seamlessly bridges neural scene reconstruction and physics simulation through a differentiable particle-based physical simulator and the proposed SP-MC that efficiently transforms implicit representations into explicit surface points. Our method enables differentiable optimization with both rendering and physical losses.

2. We propose a novel method that jointly models rendering and physical uncertainties for 3D reconstruction. By dynamically adjusting the per-pixel rendering loss and physics-guided pixel sampling, our model significantly improves the reconstruction of thin structures.

3. Extensive experiments demonstrate that our model significantly enhances reconstruction quality and physical plausibility, outperforming state-of-the-art methods. Our results exhibit substantial stability improvements, signaling broader potential for physics-demanding applications.

## 2    Related Work

**Neural Implicit Surface Reconstruction**    With the increasing popularity of neural implicit representations [48, 79, 35, 6, 49, 7, 40] in 3D reconstruction, recent studies [50, 64, 75] have bridged the volume density in Neural Radiance Fields (NeRF) [45, 80] with the iso-surface representation, *e.g.*, occupancy [42] or signed distance function (SDF) [52] to enable reconstruction from 2D images. Furthermore, advanced methods [68, 29, 69, 32] achieve compositional scene reconstruction by decomposing the latent 3D representation into the background and foreground objects. Despite achieving plausible object disentanglement, these methods fail to yield physically plausible reconstruction results, primarily due to the absence of physics constraints in existing neural implicit reconstruction pipelines. This paper addresses this limitation by incorporating both appearance and physical cues, thereby enabling surface learning with both rendering and physical losses.

**Incorporating Priors into Neural Surface Reconstruction**    Various priors, such as Manhattan-world assumptions [20, 18], monocular geometric priors (*i.e.*, normal [63] and depth [77] from off-the-shelf model [55, 12, 2]) and diffusion priors [70, 34], have been employed to improve optimization and robustness in surface reconstruction, especially in texture-less regions of indoor scenes. However, these priors primarily involve geometry and appearance, neglecting physics priors despite their critical role in assessing object shapes and achieving stability. This paper proposes to incorporate physical priors through a differentiable simulator, with physical loss to penalize object movement and a joint uncertainty modeling of both rendering and physics. Rendering uncertainty filters multi-view inconsistencies in the monocular geometric priors, akin to prior work [41, 73, 51, 71]. The physical uncertainty employs the 3D contact points from the simulator to offer accurate insights into areas lacking physical support, which helps modulate the rendering losses and guides pixel sampling.

**Physics in Scene Understanding**    There has been an increasing interest in incorporating physics commonsense in the scene understanding community, spanning various topics such as object generation [47, 14, 61, 44, 43], object decomposition [36, 82, 5], material simulation [23, 31, 72, 81], scene synthesis [54, 74], and human motion generation [57, 59, 22, 78, 66, 21, 4, 8, 25, 65, 24]. In the context of scene reconstruction, previous work primarily focuses on applying collision loss on 3D bounding boxes to adjust their translations and rotations [11, 3], and penalize the penetration between objects [79, 69]. Ma *et al.* [38] controls objects within a differentiable physical environment and integrates differentiable rendering to improve the generalizability of object control in unknown rendering configurations. In this paper, we propose to incorporate physics information into the neural implicit surface representation through a differentiable simulator, significantly enhancing both the richness of the physical information and the granularity of the optimization. The simulator not only furnishes detailed object trajectories but also yields detailed information about the object shapes in the form of 3D contact points. With physical loss directly backpropagated to the implicit shape representation, our approach leads to a refined optimization of object shapes and physical stability.

## 3    Method

Given an input set of $N$ posed RGB images $\mathcal{I} = \{I_1, \ldots, I_N\}$ and corresponding instance masks $\mathcal{S} = \{S_1, \ldots, S_N\}$, our objective is to reconstruct each object and the background in the scene. Fig. 2 presents an overview of our proposed PHYRECON.

### 3.1    Background

**Volume Rendering of SDF-based Implicit Surfaces**    We utilize neural implicit surfaces with SDF to represent 3D geometry for implicit reconstruction. The SDF provides a continuous function that yields the distance $s(\boldsymbol{p})$ to the closest surface for a given point $\boldsymbol{p}$, with the sign indicating whether

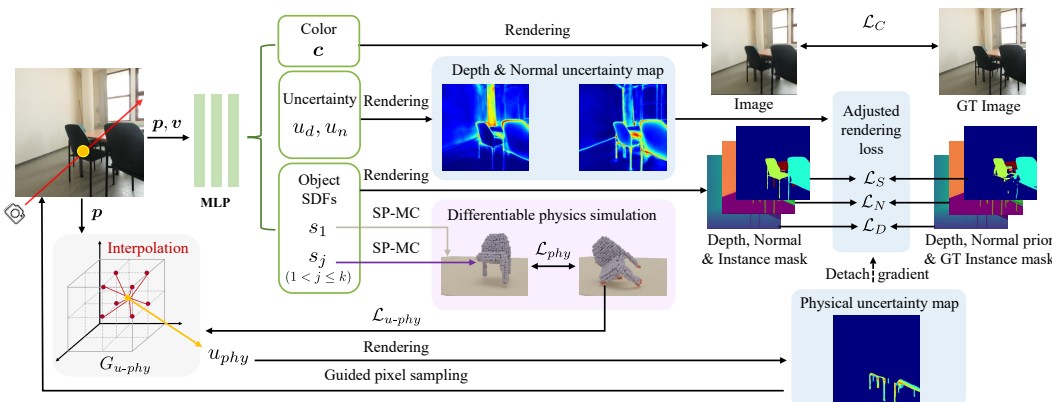

Figure 2: **Overview of PHYRECON**. We incorporate explicit physical constraints in the neural scene reconstruction through a differentiable particle-based physical simulator and a differentiable transformation (*i.e.*, SP-MC) between implicit surfaces and explicit surface points in Sec. 3.2. To learn intricate 3D structures, we introduce rendering and physical uncertainty in Sec. 3.3 to address the inconsistencies in the geometric priors and guide the pixel sampling.

the point lies inside (negative) or outside (positive) the surface. We implement the SDF function as a multi-layer perceptron (MLP) network $f(\cdot)$, and similarly, the appearance function as $g(\cdot)$.

For each camera ray $\boldsymbol{r} = (\boldsymbol{o}, \boldsymbol{v})$ with $\boldsymbol{o}$ as the ray origin and $\boldsymbol{v}$ as the viewing direction, $n$ points $\{\boldsymbol{p}_i = \boldsymbol{o} + t_i\boldsymbol{v} \mid i = 0, 1, \ldots, n-1\}$ are sampled, where $t_i$ is the distance from the sample point to the camera center. We predict the signed distance $s(\boldsymbol{p}_i)$ and the color $\boldsymbol{c}_i$ for each point along the ray, and the normal $\boldsymbol{n}_i$ is the analytical gradient of the $s(\boldsymbol{p}_i)$. The predicted color $\hat{\boldsymbol{C}}(\boldsymbol{r})$, depth $\hat{D}(\boldsymbol{r})$, and normal $\hat{\boldsymbol{N}}(\boldsymbol{r})$ for the ray $\boldsymbol{r}$ are computed with the unbiased rendering method following NeuS [64]:

$$\hat{\boldsymbol{C}}(\boldsymbol{r}) = \sum_{i=0}^{n-1} T_i \alpha_i \boldsymbol{c}_i, \quad \hat{D}(\boldsymbol{r}) = \sum_{i=0}^{n-1} T_i \alpha_i t_i, \quad \hat{\boldsymbol{N}}(\boldsymbol{r}) = \sum_{i=0}^{n-1} T_i \alpha_i \boldsymbol{n}_i, \quad (1)$$

where $T_i$ is the discrete accumulated transmittance and $\alpha_i$ is the discrete opacity value, defined as:

$$T_i = \prod_{j=0}^{i-1}(1 - \alpha_j), \quad \alpha_i = \max\left(\frac{\Phi_u(s(\boldsymbol{p}_i)) - \Phi_u(s(\boldsymbol{p}_{i+1}))}{\Phi_u(s(\boldsymbol{p}_i))}, 0\right), \quad (2)$$

where $\Phi_u(x) = (1 + e^{-ux})^{-1}$ and $u$ is a learnable parameter.

**Object-compositional Scene Reconstruction**    Following previous work [68, 69, 32], we consider the compositional reconstruction of $k$ objects utilizing their corresponding masks, and we treat the background as an object. More specifically, for a scene with $k$ objects, we predict $k$ SDFs at each point $\boldsymbol{p}$, and the $j$-th ($1 \leq j \leq k$) SDF represents the geometry of the $j$-th object. Without loss of generality, we set $j = 1$ as the background object and others as the foreground objects. In subsequent sections of the paper, we denote the $j$-th object SDF at point $\boldsymbol{p}$ as $s_j(\boldsymbol{p})$. The scene SDF $s(\boldsymbol{p})$ is the minimum of the object SDFs, *i.e.*, $s(\boldsymbol{p}) = \min s_j(\boldsymbol{p}), j = 1, 2, \ldots, k$, which is used for sampling points along the ray and volume rendering in Eqs. (1) and (2). See more details in the Appx. A.

## 3.2 Differentiable Physics Simulation in Neural Scene Reconstruction

To jointly optimize the neural implicit representations with rendering and physical losses, we propose a particle-based physical simulator and a highly efficient method to transition implicit SDF, which is adept at modeling visual appearances, to explicit representations conducive to physics simulation.

**Surface Points Marching Cubes (SP-MC)**    Existing methods [13, 43] fall short of attaining a satisfactory balance between efficiency and precision in extracting surface points from the SDF-based implicit representation. Thus, we develop a novel algorithm, SP-MC, with differentiable and efficient parallel operations and surface point refinement leveraging the SDF network $f(\cdot)$.

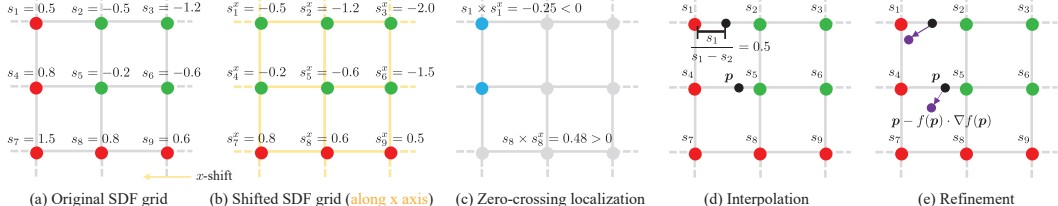

| (a) Original SDF grid | (b) Shifted SDF grid (along x axis) | (c) Zero-crossing localization | (d) Interpolation | (e) Refinement |

Figure 3: **Illustration of SP-MC**. (a-b) We first shift the SDF grids, and (c) localize the zero-crossing vertices $V$ (blue). (d) The coarse surface points $P_{\text{coarse}}$ (black) are derived through linear interpolation and (e) the fine-grained points $P_{\text{fine}}$ (purple) are obtained by querying the SDF network $f(\cdot)$.

Fig. 3 illustrates the SP-MC algorithm. Formally, let $S \in \mathbb{R}^{N \times N \times N}$ denote a signed distance grid of size $N$ obtained using $f(\cdot)$, where $S(p) \in \mathbb{R}$ denotes the signed distance of the vertex $p$ to its closest surface. SP-MC first locates the zero-crossing vertices $V$, where a sign change occurs from the neighboring vertices in signed distances, by shifting the SDF grid along the $x$, $y$, and $z$ axis, respectively. For example, shifting along the $x$ axis means $S^x(i, j, k) = S(i + 1, j, k)$. Then each vertex $p \in V$ can be found through the element-wise multiplication of $S$ and the shifted grids $S^{d \in \{x, y, z\}}$, together with its sign-flipping neighbor vertex denoted as $p^{\text{shift}}$. The coarse surface points $P_{\text{coarse}}$ are determined along the grid edge via linear interpolation:

$$P_{\text{coarse}} = \left\{ p + \frac{S(p)}{S(p) - S(p^{\text{shift}})} (p^{\text{shift}} - p) \mid p \in V \right\}. \tag{3}$$

Finally, $P_{\text{coarse}}$ are refined using their surface normals and signed distances by querying $f(\cdot)$:

$$P_{\text{fine}} = \left\{ p - f(p) \cdot \nabla f(p) \mid p \in P_{\text{coarse}} \right\}. \tag{4}$$

Benefiting from all operations that are friendly for parallel computation, SP-MC is capable of extracting object surface points faster than the Kaolin [13] algorithm with less GPU memory consumption. In the meantime, it also achieves unbiased estimation of the surface points compared with direct thresholding the signed distance field [43], which is crucial for learning fine structures. For detailed algorithm and quantitative computational comparisons, please refer to the Appx. B.

**Particle-based Physical Simulator** To incorporate physics constraints into the neural implicit reconstruction, we develop a fully differentiable particle-based simulator implemented with Diff-Taichi [19]. Our simulator is designed for realistic simulations between rigid bodies, which are depicted as a collection of spherical particles of uniform size. The simulator captures the body's dynamics through translation and rotation, anchored to a reference coordinate system. The body state at any time is given by its position, orientation, linear and angular velocities. The total mass and the moment of inertia are intrinsic physical properties of the rigid body, reflecting its resistance to external forces and torques. During forward dynamics, the states are updated through time using explicit Euler time integration. Besides gravity, the changes in the rigid body state are induced by contact and friction forces, which are resolved using an impulse-based method. For more implementation details, please refer to the Appx. C.

The simulator is used to track the trajectory and contact state of the object's surface points $P_{\text{fine}}$ until reaching stability or maximum simulation steps. Utilizing the trajectories and contact states, we present our simple yet effective physical loss: $\mathcal{L}_{phy} = \sum_{p \in P_{\text{contact}}} \left\| p' - p^0 \right\|_2$ , where the initial position of each point $p$ is denoted as $p^0$ and its first contact position with the supporting plane as $p'$. Our physical loss intuitively penalizes the object's movement and is only applied to the contact points $P_{\text{contact}}$ instead of all the surface points. If the physical loss is homogeneously applied to all the surface points, it will contradict the rendering cues when the object is unstable, leading to degenerated results. The contact points $P_{\text{contact}}$, on the other hand, indicate the areas of object instability. As shown in the experiment section, our design leads to stable 3D geometry under the coordination between the appearance and physics constraints.

## 3.3 Joint Uncertainty Modeling

Apart from incorporating explicit physical constraints, we propose a joint uncertainty modeling approach, encompassing both rendering and physical uncertainty, to mitigate the inconsistencies and improve the reconstruction of thin structures, which is crucial for object stability.

**Rendering Uncertainty** The rendering uncertainty is designed to address the issue of multi-view and multi-modal inconsistencies in monocular geometry priors, including depth uncertainty and normal uncertainty. We model the depth uncertainty $u_d$ and normal uncertainty $u_n$ of a 3D point as view-dependent representations [51, 58, 71], which we utilize the appearance network $g(\cdot)$ to predict along with the color $\boldsymbol{c}$ for a 3D point $\boldsymbol{p}$:

$$g : (\boldsymbol{p} \in \mathbb{R}^3, \boldsymbol{n} \in \mathbb{R}^3, \boldsymbol{v} \in \mathbb{R}^3, \boldsymbol{f} \in \mathbb{R}^{256}) \mapsto (\boldsymbol{c} \in \mathbb{R}^3, u_d \in \mathbb{R}, u_n \in \mathbb{R}), \quad (5)$$

where $\boldsymbol{n}$ is the normal at $\boldsymbol{p}$, $\boldsymbol{v}$ is the viewing direction, and $\boldsymbol{f}$ is a geometry latent feature output from the SDF network $f(\cdot)$.

The depth uncertainty $\hat{U}_d(\boldsymbol{r})$ and normal uncertainty $\hat{U}_n(\boldsymbol{r})$ of ray $\boldsymbol{r}$ are computed using $u_d$ and $u_n$ through volume rendering, similar to Eq. (1). Subsequently, we modulate the $L_2$ depth loss $\mathcal{L}_D(\boldsymbol{r})$ and $L_1$ normal loss $\mathcal{L}_N(\boldsymbol{r})$ based on the rendering uncertainty:

$$\mathcal{L}_{D_U}(\boldsymbol{r}) = \ln(|\hat{U}_d(\boldsymbol{r})| + 1) + \frac{\mathcal{L}_D(\boldsymbol{r})}{|\hat{U}_d(\boldsymbol{r})|}, \quad \mathcal{L}_{N_U}(\boldsymbol{r}) = \ln(|\hat{U}_n(\boldsymbol{r})| + 1) + \frac{\mathcal{L}_N(\boldsymbol{r})}{|\hat{U}_n(\boldsymbol{r})|}. \quad (6)$$

Intuitively, the uncertainty-aware rendering loss can assign higher uncertainty to the pixels with larger loss, thus filtering the inconsistent monocular priors when the prediction from one viewpoint differs from others. Fig. 6 depicts how higher uncertainty precisely localizes inconsistent monocular priors.

**Physical Uncertainty** We introduce physical uncertainty to capture object instability from physics simulations. Specifically, we represent the physical uncertainty field with a dense grid $\boldsymbol{G}_{u\text{-}phy} \in \mathbb{R}^{N \times N \times N}$. The physical uncertainty $u_{phy}(\boldsymbol{p}) \in \mathbb{R}$ for any 3D point $\boldsymbol{p}$ is obtained through trilinear interpolation. $\boldsymbol{G}_{u\text{-}phy}$ is initialized to zero and updated after each simulation trial using the 3D contact points $\boldsymbol{P}_{\text{contact}}$. For each contact point, we track its trajectory from $\boldsymbol{p}^0$ to $\boldsymbol{p}'$, forming the uncertain point set $\boldsymbol{P}_{\text{u}}$. We design a loss function $\mathcal{L}_{u\text{-}phy}$ to update $\boldsymbol{G}_{u\text{-}phy}$:

$$\mathcal{L}_{u\text{-}phy} = -\xi \sum_{\boldsymbol{p} \in \boldsymbol{P}_{\text{u}}} u_{phy}(\boldsymbol{p}), \quad (7)$$

where $\xi > 0$ represents the increasing rate for $\boldsymbol{G}_{u\text{-}phy}$. Thus, areas with high physical uncertainty indicate that the object lacks support and requires the development of supporting structures. The physical uncertainty $\hat{U}_{phy}(\boldsymbol{r})$ of the pixel corresponding to ray $\boldsymbol{r}$ is computed via volume rendering.

Finally, we present the rendering losses re-weighted by both rendering and physical uncertainty:

$$\mathcal{L}_D = \sum_{\boldsymbol{r} \in \mathcal{R}} \frac{\mathcal{L}_{D_U}(\boldsymbol{r})}{\hat{U}_{phy}(\boldsymbol{r}) + 1}, \quad \mathcal{L}_N = \sum_{\boldsymbol{r} \in \mathcal{R}} \frac{\mathcal{L}_{N_U}(\boldsymbol{r})}{\hat{U}_{phy}(\boldsymbol{r}) + 1}, \quad \mathcal{L}_S = \sum_{\boldsymbol{r} \in \mathcal{R}} \frac{\mathcal{L}_S(\boldsymbol{r})}{\hat{U}_{phy}(\boldsymbol{r}) + 1}. \quad (8)$$

The segmentation loss $\mathcal{L}_S(\boldsymbol{r})$ is weighted by physical uncertainty only since we use view-consistent instance masks following prior work [68, 69, 32]. We detach $\hat{U}_{phy}(\boldsymbol{r})$ in the above losses, ensuring the physical uncertainty field $\boldsymbol{G}_{u\text{-}phy}$ reflects accurate physical information from the simulator.

**Physics-Guided Pixel Sampling** A critical observation is that intricate object structures occupy only a minor portion of the image, leading to a lower probability of being sampled despite their significance in preserving object stability. Effectively pinpointing these fine geometries is perceptually challenging, yet physical uncertainty precisely outlines the areas where the object lacks support and requires special attention for optimization. Hence, we introduce a physics-guided pixel sampling strategy to enhance the learning of detailed 3D structures. Using physical uncertainty, we calculate the sampling probability of each pixel in the entire image by: $p(\boldsymbol{r}) = \dfrac{\hat{U}_{phy}(\boldsymbol{r})}{\sum_{\boldsymbol{r} \in \mathcal{R}} \hat{U}_{phy}(\boldsymbol{r})}$.

## 3.4 Training Details

In summary, our overall loss function is: $\mathcal{L} = \mathcal{L}_{RGB} + \mathcal{L}_D + \mathcal{L}_N + \mathcal{L}_S + \mathcal{L}_{phy} + \mathcal{L}_{u\text{-}phy} + \mathcal{L}_{reg}$, where the loss weights are omitted for simplicity. Following prior work [16, 77, 32], we introduce

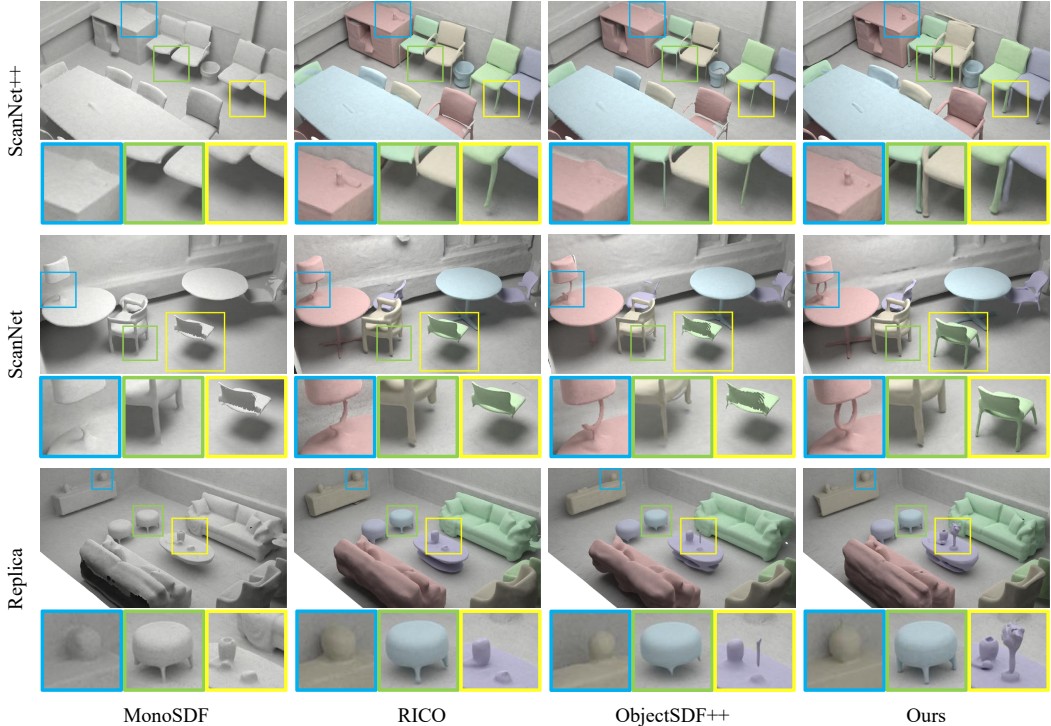

Figure 4: **Qualitative results of indoor scene reconstruction.** Examples from ScanNet++ [76], ScanNet [9], and Replica [62] demonstrate that our model produces higher quality reconstructions compared to the baselines. Our results contain finer details for slender structures (*e.g.*, chair legs and objects on the table) and plausible support relations, which are highlighted in the zoom-in boxes.

$\mathcal{L}_{reg}$ to regularize the unobservable regions for background and foreground objects, as well as the implicit shapes through an eikonal term.

To ensure robust optimization and coordination among various components, we empirically divide the training into three stages. In the first stage, we exclusively leverage the rendering losses with rendering uncertainties. In the second stage, we introduce our physical simulator to incorporate physical uncertainty into rendering losses and enable physics-guided pixel sampling. Finally, we integrate the physical loss using a learning curriculum that gradually increases the physical loss weight. The staged training allows the simulator to pinpoint more accurate contact points, essential for effective optimization of shape through the physical loss. For further details about the loss, training, and implementation details, please refer to Appx. A.

## 4 Experiments

We assess the effectiveness of PHYRECON by evaluating scene and object reconstruction quality, as well as object stability. We present additional results in Appx. D and limitations in Appx. E.

### 4.1 Settings

**Datasets** We conduct experiments on both the synthetic dataset Replica [62] and real datasets ScanNet [9] and ScanNet++ [76]. We use 8 scenes from Replica following the setting of MonoSDF [77] and ObjectSDF++ [69]. For ScanNet, we use the 7 scenes following RICO [32]. Additionally, we conduct experiments on 7 scenes from the more recent real-world dataset ScanNet++ with higher-quality images and more accurate camera poses. More data preparation details are in Appx. D.1.

**Baselines and Metrics** We choose MonoSDF [77], RICO [32], and ObjectSDF++ [69] as the baseline models. For reconstruction, we measure the Chamfer Distance (CD), the F-score (F-score),

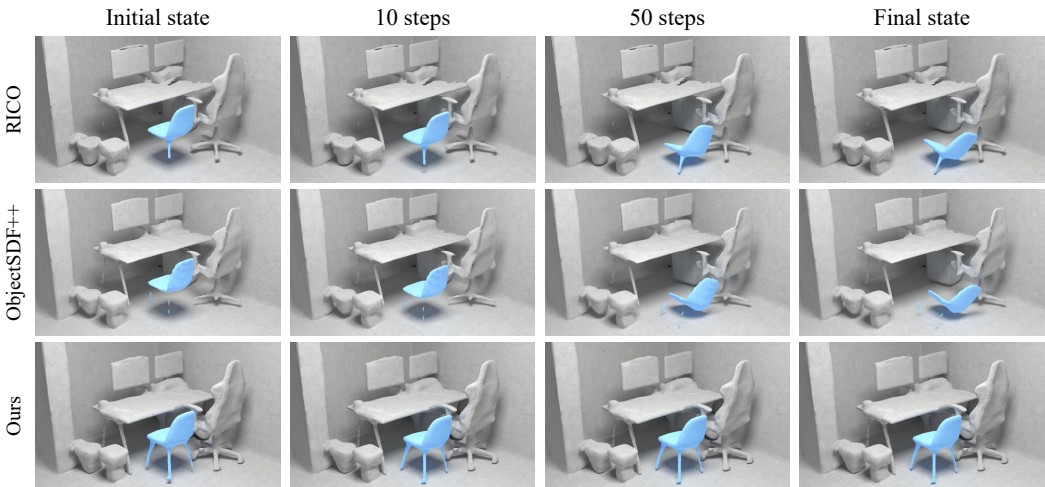

Figure 5: **Object trajectory during simulation.** Our method enhances the physical plausibility of the reconstruction results, which can remain stable during dropping simulation in Isaac Gym.

Table 1: **Quantitative results of 3D reconstruction.** Our model reaches the best reconstruction quality across all datasets and significantly improves on physical stability.

| Dataset | Method | Scene Recon. | | | Obj. Recon. | | | Obj. Stability |
|---------|--------|------|---------|------|------|---------|------|----------------|
| | | CD↓ | F-Score↑ | NC↑ | CD↓ | F-score↑ | NC↑ | SR (%) ↑ |
| ScanNet++ | MonoSDF | 3.94 | 78.14 | 89.37 | - | - | - | - |
| | RICO | 3.87 | 78.45 | 89.64 | 4.29 | 85.91 | 85.45 | 26.43 |
| | ObjectSDF++ | 3.79 | 79.12 | 89.57 | 4.08 | 86.32 | 85.32 | 25.28 |
| | Ours | **3.34** | **81.53** | **90.10** | **3.28** | **87.21** | **86.16** | **78.16** |
| ScanNet | MonoSDF | 8.97 | 60.30 | 84.40 | - | - | - | - |
| | RICO | 8.92 | 61.44 | 84.58 | 9.29 | 73.10 | 79.44 | 29.68 |
| | ObjectSDF++ | 8.86 | 61.68 | 85.20 | 9.21 | 74.82 | 81.05 | 26.56 |
| | Ours | **8.34** | **63.01** | **86.57** | **7.92** | **75.54** | **82.54** | **70.31** |
| Replica | MonoSDF | 3.87 | 85.01 | 88.59 | - | - | - | - |
| | RICO | 3.86 | 84.66 | 88.68 | 4.16 | 80.38 | 84.30 | 32.89 |
| | ObjectSDF++ | 3.73 | 85.50 | 88.60 | 3.99 | 80.71 | 84.22 | 30.26 |
| | Ours | **3.68** | **85.61** | **89.45** | **3.86** | **81.30** | **84.91** | **77.63** |

and Normal Consistency (NC) following MonoSDF [77]. These metrics are evaluated for both scene and object reconstruction across all datasets. As MonoSDF is designed to reconstruct the whole scene, we only evaluate its scene reconstruction quality. To evaluate the physical plausibility of the reconstructed objects, we report the Stability Ratio (SR), defined as the ratio of the number of physically stable objects against the total number of objects in the scene. The assessment is conducted using dropping simulation via the Isaac Gym simulator [39] to avoid bias in favor of our method. We provide more details on our evaluation procedure in Appx. D.2.

## 4.2 Results

Tab. 1 and Fig. 4 present the quantitative and qualitative results of the neural scene reconstruction. In Fig. 5, we visualize the trajectory for the reconstructed object during dropping simulation in Isaac Gym [39]. Fig. 6 showcases visualizations of several critical components in our methods, including the sampling strategy, volume renderings of the geometric cues, and the uncertainty maps. Our method significantly outperforms all baselines, and we summarize the key observations as follows.

**Physical Stability** From the results in Tab. 1, our method realizes significant improvements in object stability, outperforming all baselines by at least 40% across all datasets. This signifies *a significant*

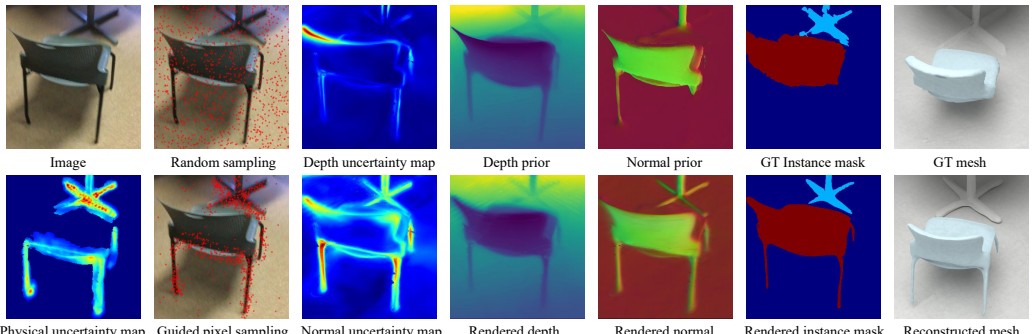

Figure 6: **Joint uncertainty modeling.** The physical uncertainty pinpoints the regions critical for stability, efficiently guiding the pixel sampling. The rendering uncertainties can alleviate the impact of inconsistent geometry cues, leading to a better-reconstructed mesh than the GT.

*stride towards achieving physically plausible scene reconstruction* from multi-view images. The notable stability improvement is attributed to two essential factors: 1) the integration of physical loss and physical uncertainty ensures the reconstructed structure closely aligns with the ground floor, as evident in the examples from Figs. 4 and 5 and 2) enhanced learning of intricate structures, facilitated by joint uncertainty modeling and physics-guided pixel sampling. Results from RICO [32] and ObjectSDF++ [69] in Fig. 5 fail to capture all the chair legs, leading to inherent instability.

**Reconstruction Quality**    Note that the improvement of the physical plausibility does not come under the sacrifice of reconstruction quality. From the results in Tab. 1, our method surpasses all the state-of-the-art methods across three datasets in all the reconstruction metrics. As also shown in Fig. 4, while the baseline methods are capable of reconstructing substantial parts of objects, *e.g.*, sofa or tabletop, they struggle with the intricate structures of objects. In contrast, our model achieves much more detailed reconstruction, *e.g.*, the vase and lamp on the tables, shown in the zoom-in views.

**Joint Uncertainty Modeling**    Lastly, we discuss the intermediate results of our joint uncertainty modeling exemplified on the ScanNet dataset. As shown in Fig. 6, the physical uncertainty adeptly pinpoints the regions critical for remaining stability, such as the chair legs and table base. The physics-guided pixel sampling, informed by the physical uncertainty map, prioritizes intricate structures over the random sampling strategy. Moreover, it modulates rendering losses, particularly useful for instance mask loss where chair legs are absent in the ground-truth instance mask. Meanwhile, the depth and normal uncertainty maps identify inconsistencies in the geometric prior, *e.g.*, miss detections in the normal prior and the overly sharp depth prior. Collectively, they contribute to the physically plausible and detailed reconstructed mesh, surpassing the ground truth.

Table 2: **Ablation results on ScanNet++ dataset.**

| $RU$ | $PU$ | $PS$ | $PL$ | Scene Recon. | | | Obj. Recon. | | | Obj. Stability |
|---|---|---|---|---|---|---|---|---|---|---|
| | | | | CD↓ | F-Score↑ | NC↑ | CD↓ | F-score↑ | NC↑ | SR (%) ↑ |
| × | × | × | × | 3.82 | 78.64 | 89.52 | 4.13 | 86.26 | 85.42 | 25.28 |
| × | × | × | ✓ | 3.68 | 79.35 | 89.55 | 3.97 | 86.35 | 85.26 | 68.96 |
| ✓ | × | × | × | 3.46 | 80.73 | 89.63 | 3.65 | 86.83 | 85.53 | 47.12 |
| ✓ | ✓ | × | × | 3.42 | 80.68 | 89.67 | 3.46 | 86.97 | 85.45 | 56.32 |
| ✓ | ✓ | ✓ | × | **3.31** | **81.64** | 89.94 | 3.34 | 87.17 | 85.47 | 60.91 |
| ✓ | ✓ | ✓ | ✓ | 3.34 | 81.53 | **90.10** | **3.28** | **87.21** | **86.16** | **78.16** |

## 4.3   Ablation Study

We conduct ablative studies on the physical loss ($PL$), rendering uncertainty ($RU$), physical uncertainty ($PU$), and physics-guided pixel sampling ($PS$). Tab. 2 and Fig. 7 illustrate quantitative and qualitative comparisons, respectively. Key findings are as follows:

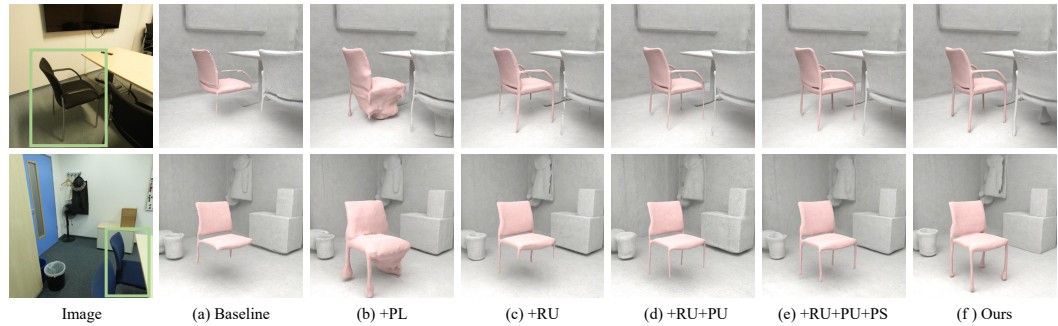

| Image | (a) Baseline | (b) +PL | (c) +RU | (d) +RU+PU | (e) +RU+PU+PS | (f) Ours |

Figure 7: **Visual comparisons for ablation study.** $PL$ denotes physical loss, $RU$ for rendering uncertainty, $PU$ for physical uncertainty and $PS$ for physics-guided sampling.

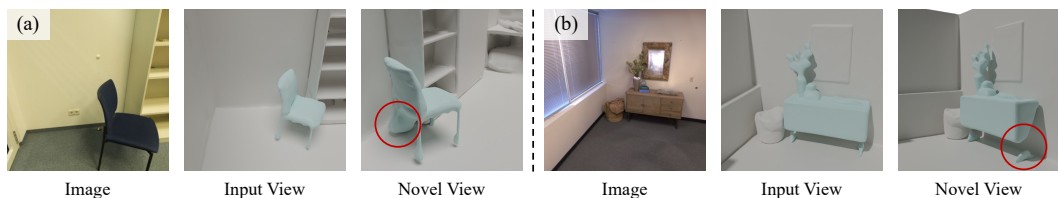

| Image | Input View | Novel View | Image | Input View | Novel View |

Figure 8: **Qualitative examples for failure cases.**

1. Physical loss significantly improves 3D object stability. However, due to the insufficient regularization of thin structures and imprecise contact points in the simulation, adding physical loss alone will overshadow the rendering losses, leading to degenerated object shapes to sustain stability.
2. Rendering uncertainty, physical uncertainty, and physic-guided pixel sampling collectively contribute to enhancing the reconstruction quality, particularly on thin structures. However, despite the advancements, the reconstructed results still struggle to maintain physical plausibility without direct physical supervision during the optimization.
3. When all components are introduced, the enhanced reconstruction of thin structures leads to more meaningful contact points in simulation, thus enabling effective joint optimization with physical loss. This not only improves the objects' stability but also preserves their reasonable shapes, leading to robust reconstruction with both physical plausibility and fine details simultaneously.

### 4.4 Failure Cases

We present and diagnose representative failure examples. Fig. 8 (a) demonstrates that in regions scarcely observed in the input images, optimizing with the physical loss may lead to degenerated object shapes, *e.g.*, bulges, to maintain physical stability. This is due to insufficient supervision from the rendering losses. Fig. 8 (b) illustrates that objects may be divided into several disconnected parts, which also appear stable in the simulation. This is a common limitation of the current neural reconstruction pipeline and may be addressed by incorporating topological regularization to penalize disconnected parts of the object or further enhance the physical simulator.

## 5 Conclusion

In conclusion, this paper introduces PHYRECON as the first approach to leverage both differentiable rendering and differentiable physics simulation for learning implicit surface representations. Our framework features a novel differentiable particle-based physical simulator and joint uncertainty modeling, facilitating efficient optimization with both rendering and physical losses. Extensive experiments validate the effectiveness of PHYRECON, showcasing its significant outperformance of all state-of-the-art methods in terms of both reconstruction quality and physics stability, underscoring its potential for future physics-demanding applications.

## Acknowledgement

The authors thank the anonymous reviewers for their constructive feedback. Y. Zhu is supported in part by the National Science and Technology Major Project (2022ZD0114900), the National Natural Science Foundation of China (62376031), the Beijing Nova Program, the State Key Lab of General AI at Peking University, and the PKU-BingJi Joint Laboratory for Artificial Intelligence.

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

## Appendix

In Appx. A, we first delineate the comprehensive PhyRecon algorithm, followed by providing details about the loss functions and implementation. Subsequently, we delve into the SP-MC Algorithm in Appx. B, offering a detailed comparison with other methods for extracting surface points. Next, we describe our differentiable particle-based rigid body simulator in Appx. C, covering both its theoretical derivation and concrete implementation. We also provide additional experimental details, including data preparation and evaluation metrics in Appx. D. Finally, we discuss the limitations in Appx. E, along with the potential negative impacts of our research in Appx. F. For a comprehensive understanding of the qualitative results and stability comparisons, we recommend viewing the supplementary video with detailed visualizations and animations.

## A    Model Details

### A.1    PHYRECON Overview

In Alg. 1, we illustrate the process of one training iteration of PHYRECON. Below, we provide further explanation.

First, we sample pixels from an image guided by physical uncertainty to enhance the sampling probability of slender structures of objects. Then, we sample points along a ray according to NeuS [64]. For each point, we predict individual object SDFs $\{s_1, s_2, ..., s_k\}$, color $\boldsymbol{c}$, depth uncertainty $u_d$, and normal uncertainty $u_n$ using $f(\cdot)$ and $g(\cdot)$, and obtain physical uncertainty from the physical uncertainty grid $\boldsymbol{G}_{u\text{-}phy}$ through trilinear interpolation. Subsequently, we compute color $\hat{\boldsymbol{C}}$, depth $\hat{D}$, normal $\hat{\boldsymbol{N}}$, instance logits $\hat{\boldsymbol{H}}$, depth uncertainty $\hat{U}_d$, normal uncertainty $\hat{U}_n$, and physical uncertainty $\hat{U}_{phy}$ for each ray via volume rendering [10]. In particular, the physical uncertainty of the pixel corresponding to ray $r$ is computed via:

$$\hat{U}_{phy}(\boldsymbol{r}) = \sum_{i=0}^{n-1} T_i \alpha_i u_{phy,i}. \tag{A1}$$

Note that the loop for each ray and each point here is parallelized during actual computation.

For each foreground object, we extract its surface points $\boldsymbol{P}^{\text{obj}}$ and background surface points $\boldsymbol{P}^{\text{bg}}$ using SP-MC. Utilizing our proposed physical simulator, we track the trajectories and contact states of the object points. For each point $\boldsymbol{p} \in \boldsymbol{P}^{\text{obj}}$, we identify its initial position as $\boldsymbol{p}^0$ and its first contact position with the supporting plane as $\boldsymbol{p}'$. The object points that have made contact with the supporting plane are collectively denoted as $\boldsymbol{P}_{\text{contact}}$. If a surface point $\boldsymbol{p}$ does not contact the supporting plane, $\boldsymbol{p}'$ remains equal to $\boldsymbol{p}^0$. The initial and contact positions of the contact points are used to compute the physical loss $\mathcal{L}_{phy}$, which penalizes object movement in the simulator.

To update the physical uncertainty grid $\boldsymbol{G}_{u\text{-}phy}$, we construct the uncertain point set $\boldsymbol{P}_u$ by interpolating from the initial position $\boldsymbol{p}^0$ to the first contact position $\boldsymbol{p}'$ for all the points in $\boldsymbol{P}_{\text{contact}}$. The physical uncertainty grid is optimized through $\mathcal{L}_{u\text{-}phy}$. Note we only record the essential information and do not output the full trajectory of all object points during simulations to reduce CUDA memory usage.

Finally, we compute the depth, normal, and instance semantic loss, re-weighted by both rendering and physical uncertainty using Eq. (15) from the main paper. Additionally, we calculate the color loss and regularization loss terms. Finally, we optimize the network $f(\cdot)$, $g(\cdot)$, and the physical uncertainty grid $\boldsymbol{G}_{u\text{-}phy}$ using the aforementioned losses.

### A.2    Loss Function Details

**RGB Reconstruction Loss**    To learn the surface from images input, we need to minimize the difference between ground-truth pixel color and the rendered color. We follow the previous work [77] here for the RGB reconstruction loss:

$$\mathcal{L}_{RGB} = \sum_{\boldsymbol{r} \in \mathcal{R}} ||\hat{\boldsymbol{C}}(\boldsymbol{r}) - \boldsymbol{C}(\boldsymbol{r})||_1, \tag{A2}$$

---

**Algorithm 1** PhyRecon per Training Iteration

---

1: Physics-Guided Pixel Sampling $\rightarrow \{\boldsymbol{r}\}$         *# Guided by physical uncertainty*
2: **For** *each $\boldsymbol{r}$* **do**
3:      NeuS ray sampling $\rightarrow \{\boldsymbol{p}\}$
4:      **For** *each $\boldsymbol{p}$* **do**
5:         // SDF prediction
6:         $s_1, s_2, ..., s_k, \boldsymbol{z} = f(\boldsymbol{p})$
7:         $s = \min(s_1, s_2, ..., s_k)$
8:         $\sigma = \Phi(s)$               *# $\Phi$: transform SDF to density*
9:         // Appearance and rendering uncertainty prediction
10:         $\boldsymbol{c}, u_d, u_n = g(\boldsymbol{p}, \boldsymbol{n}, \boldsymbol{v}, \boldsymbol{z})$
11:         // Physical uncertainty via trilinear interpolation
12:         $u_{phy} = \text{Interp}(\boldsymbol{G}_{u\text{-}phy}, \boldsymbol{p})$
13:      **end for**
14:      $\hat{\boldsymbol{C}}, \hat{D}, \hat{\boldsymbol{N}}, \hat{\boldsymbol{H}}, \hat{U}_d, \hat{U}_n, \hat{U}_{\text{phy}} = \text{VolumeRendering}(\cdot)$
15: **end for**
16: **For** *j=2,...,k* **do**
17:      // Background surface points extraction
18:      $\boldsymbol{P}^{\text{bg}} = \text{SP-MC}(\boldsymbol{s}_1)$
19:      // Object surface points extraction
20:      $\boldsymbol{P}^{\text{obj}} = \text{SP-MC}(\boldsymbol{s}_j)$
21:      // Differentiable Physics Simulation
22:      $\{\boldsymbol{p}^0, \boldsymbol{p}'\}, \boldsymbol{P}_{\text{contact}} = \text{Simulator}(\boldsymbol{P}^{\text{obj}}, \boldsymbol{P}^{\text{bg}})$    *# $\boldsymbol{p}^0$: initial position, $\boldsymbol{p}'$: first contact position*
23:      // Physical uncertainty points extraction
24:      $\boldsymbol{P}_{\text{u}} = \sum_{\boldsymbol{p} \in \boldsymbol{P}_{\text{contact}}} \text{Interp}(\boldsymbol{p}^0, \boldsymbol{p}')$
25:      // Physics-related losses
26:      $L_{u\text{-}phy} = -\xi \sum_{\boldsymbol{p} \in \boldsymbol{P}_{\text{u}}} u_{phy}(\boldsymbol{p})$
27:      $L_{phy} = \sum_{\boldsymbol{p} \in \boldsymbol{P}_{\text{contact}}} L_1(\boldsymbol{p}^0, \boldsymbol{p}')$
28: **end for**
29: // Rendering losses modulated by $\hat{U}_d, \hat{U}_n, \hat{U}_{phy}$
30: Compute $L_c, L_d, L_n, L_s, L_{reg}$
31: Optimize network $f(\cdot), g(\cdot)$ and physical uncertainty grid $\boldsymbol{G}_{u\text{-}phy}$

---

where $\hat{\boldsymbol{C}}(\boldsymbol{r})$ is the rendered color from volume rendering and $\boldsymbol{C}(\boldsymbol{r})$ denotes the ground truth.

**Depth Consistency Loss**    Monocular depth and normal cues [77] can greatly benefit indoor scene reconstruction. For the depth consistency, we minimize the difference between rendered depth $\hat{D}(\boldsymbol{r})$ and the depth estimation $\bar{D}(\boldsymbol{r})$ from the Marigold model [27]:

$$\mathcal{L}_D = \sum_{\boldsymbol{r} \in \mathcal{R}} ||(w\hat{D}(\boldsymbol{r}) + q) - \bar{D}(\boldsymbol{r})||^2, \tag{A3}$$

where $w$ and $q$ are the scale and shift values to match the different scales. We solve $w$ and $q$ with a least-squares criterion, which has the closed-form solution. Please refer to the supplementary material of [77] for a detailed computation process.

**Normal Consistency Loss**    We utilize the normal cues $\bar{\boldsymbol{N}}(\boldsymbol{r})$ from Omnidata model [12] to supervise the rendered normal through the normal consistency loss, which comprises L1 and angular losses:

$$\mathcal{L}_N = \sum_{\boldsymbol{r} \in \mathcal{R}} ||\hat{\boldsymbol{N}}(\boldsymbol{r}) - \bar{\boldsymbol{N}}(\boldsymbol{r})||_1 + ||1 - \hat{\boldsymbol{N}}(\boldsymbol{r})^T \bar{\boldsymbol{N}}(\boldsymbol{r})||_1. \tag{A4}$$

The volume-rendered normal and normal estimation will be transformed into the same coordinate system by the camera pose.

**Semantic Loss**    Building on previous work [68, 32], we transform each point's SDFs into instance logits $\boldsymbol{h}(\boldsymbol{p}) = [h_1(\boldsymbol{p}), h_2(\boldsymbol{p}), \ldots, h_k(\boldsymbol{p})]$, where

$$h_j(\boldsymbol{p}) = \gamma/(1 + \exp(\gamma \cdot s_j(\boldsymbol{p}))). \tag{A5}$$

Here, $\gamma$ is a fixed parameter. Subsequently, we can obtain the instance logits $\hat{\boldsymbol{H}}(\boldsymbol{r}) \in \mathbb{R}^k$ of pixel corresponds to the ray $\boldsymbol{r}$ using volume rendering as:

$$\hat{\boldsymbol{H}}(\boldsymbol{r}) = \sum_{i=0}^{n-1} T_i \alpha_i \boldsymbol{h}_i. \tag{A6}$$

We minimize the semantic loss between volume-rendered semantic logits of each pixel and the ground-truth pixel semantic class. The semantic objective is implemented as a cross-entropy loss:

$$\mathcal{L}_S = \sum_{\boldsymbol{r} \in \mathcal{R}} \sum_{j=1}^{k} -\bar{h}_j(\boldsymbol{r}) \log h_j(\boldsymbol{r}). \tag{A7}$$

The $\bar{h}_j(\boldsymbol{r})$ is the ground-truth semantic probability for $j$-th object, which is 1 or 0.

**Eikonal Loss**   Following common practice, we also add an Eikonal term on the sampled points to regularize the SDF learning by:

$$\mathcal{L}_{Eikonal} = \sum_i^n (\|\nabla \min_{1 \le j \le k} s_j(\boldsymbol{p}_i)\|_2 - 1). \tag{A8}$$

The Eikonal loss is applied to the gradient of the scene SDF, which is the minimum of all the SDFs.

**Background Smoothness Loss**   Building upon RICO [32], we use background smoothness loss to regularize the geometry of the occluded background to be smooth. Specifically, we randomly sample a $\mathcal{P} \times \mathcal{P}$ size patch every $\mathcal{T}_\mathcal{P}$ iterations within the given image and compute semantic map $\hat{\boldsymbol{H}}(\boldsymbol{r})$ and a patch mask $\hat{M}(\boldsymbol{r})$:

$$\hat{M}(\boldsymbol{r}) = \mathbb{1}[\arg\max(\hat{\boldsymbol{H}}(\boldsymbol{r})) \ne 1], \tag{A9}$$

wherein the mask value is 1 if the rendered class is not the background, thereby ensuring only the occluded background is regulated. Subsequently, we calculate the background depth map $\hat{D}(\boldsymbol{r})$ and background normal map $\hat{\boldsymbol{N}}(\boldsymbol{r})$ using the background SDF exclusively. The patch-based background smoothness loss is then computed as:

$$\mathcal{L}(\hat{D}) = \sum_{d=0}^{3} \sum_{m,n=0}^{\mathcal{P}-1-2^d} \hat{M}(\boldsymbol{r}_{m,n}) \odot (|\hat{D}(\boldsymbol{r}_{m,n}) - \hat{D}(\boldsymbol{r}_{m,n+2^d})| + |\hat{D}(\boldsymbol{r}_{m,n}) - \hat{D}(\boldsymbol{r}_{m+2^d,n})|),$$

$$\mathcal{L}(\hat{\boldsymbol{N}}) = \sum_{d=0}^{3} \sum_{m,n=0}^{\mathcal{P}-1-2^d} \hat{M}(\boldsymbol{r}_{m,n}) \odot (|\hat{\boldsymbol{N}}(\boldsymbol{r}_{m,n}) - \hat{\boldsymbol{N}}(\boldsymbol{r}_{m,n+2^d})| + |\hat{\boldsymbol{N}}(\boldsymbol{r}_{m,n}) - \hat{\boldsymbol{N}}(\boldsymbol{r}_{m+2^d,n})|),$$

$$\tag{A10}$$

$$\mathcal{L}_{bs} = \mathcal{L}(\hat{D}) + \mathcal{L}(\hat{\boldsymbol{N}}) \tag{A11}$$

**Object Point-SDF Loss and Reversed Depth Loss**   To regularize SDFs of the object and the background, we further employ an object point-SDF loss to regulate objects within the room $\mathcal{L}_{op}$ and a reversed depth loss $\mathcal{L}_{rd}$, following previous work [32].

Specifically, for the sampled points along the rays, we initially apply a root-finding algorithm among the background SDF of these points to determine the zero-SDF ray depth $t'$. Then, the object point-SDF loss can be expressed as:

$$\mathcal{L}_{op} = \frac{1}{k-1} \sum_{j=2}^{k} \max(0, \epsilon - s_j(\boldsymbol{p}(t_i))) \cdot \mathbb{1}[t_i > t'], \tag{A12}$$

which pushes the objects' SDFs at points behind the surface to be greater than a positive threshold $\epsilon$.

Moreover, $\mathcal{L}_{rd}$ optimizes the entire ray's SDF distribution rather than focusing solely on discrete points. Specifically, the ray depths $\{t_i | i = 0, 1, \ldots, n-1\}$ are transformed into the reversed ray depths, denoted as $\{\hat{t}_i | i = 0, 1, \ldots, n-1\}$, where

$$\hat{t}_i = (t_0 + t_{n-1}) - t_{n-1-i}. \tag{A13}$$

The reversed depth $d_o$ of the hitting object is determined by the pixel's rendered semantic and $d_b$ of the background. The reversed depth loss is computed as:

$$\mathcal{L}_{rd} = \max(0, d_b - d_o), \tag{A14}$$

Finally, the regularization loss in our main paper $\mathcal{L}_{reg}$ is computed as follows:

$$\mathcal{L}_{reg} = \mathcal{L}_{bs} + \mathcal{L}_{op} + \mathcal{L}_{rd} + \mathcal{L}_{Eikonal} \tag{A15}$$

### A.3   Implementation Details

We implement our model in PyTorch [53] and utilize the Adam optimizer [28] with an initial learning rate of $5e - 4$. We sample 1024 rays per iteration. When incorporating physics-guided pixel sampling, we allocate 768 rays for physics-guided pixel sampling and the remaining 256 rays for random sampling. Our model is trained for 450 epochs on ScanNet [9] and ScanNet++ [76] datasets, and 2000 epochs on Replica [62] dataset. As introduced in Sec. 3.4, training is divided into three stages. For the ScanNet [9] and ScanNet++ [76] datasets, the second and final stages begin at the $360^{th}$ and $430^{th}$ epochs, respectively, while for the Replica [62] dataset, these stages start at $1700^{th}$ and $1980^{th}$ epochs. All experiments are conducted on a single NVIDIA-A100 GPU.

Following previous work [77, 32], we set 1, 0.1, 0.05, 0.04, 0.05, 0.1, 0.1 and 0.1 as loss weights for $\mathcal{L}_{RGB}$, $\mathcal{L}_D$, $\mathcal{L}_N$, $\mathcal{L}_S$, $\mathcal{L}_{Eikonal}$, $\mathcal{L}_{bs}$, $\mathcal{L}_{op}$, $\mathcal{L}_{rd}$, respectively. Additionally, we set $\xi = 100$ for updating $\boldsymbol{G}_{u\text{-}phy}$, initialize the loss weight for $\mathcal{L}_{phy}$ as 60, and increase it by 30 per epoch.

## B   Surface Points Marching Cubes (SP-MC)

Integrating a physical simulator into the learning of the SDF-based implicit representation demands highly efficient and accurate extraction of surface points. The SP-MC algorithm is inspired by the marching cube algorithm [37] which estimates the topology (*i.e.*, the vertices and connectivity of triangles) in each cell of the volumetric grid. Since surface points are only required for simulation, we improve the operation efficiency and combine the implicit SDF network $f(\cdot)$ to create fine-grained surface points.

### B.1   SP-MC Algorithm Details

The Surface Points Marching Cubes (SP-MC) is divided into three steps for extracting surface points from SDF-based implicit surface representation. First, the object is voxelized to obtain a discretized signed distance field $\boldsymbol{S} \in \mathbb{R}^{N \times N \times N}$ with grid vertices denoted as $\boldsymbol{P}$. Second, we shift the SDF grid $\boldsymbol{S}$ along the $x$, $y$, and $z$ axis, respectively, to locate zero-crossing vertices. For example, shifting along the $x$ axis results in $\boldsymbol{S}^x(i, j, k) = \boldsymbol{S}(i + 1, j, k)$. We then obtain coarse surface points $\boldsymbol{P}_{\text{coarse}}$ through linear interpolation. We refer to this step as the *Shift-Interpolation operation* in Alg. 2. Third, $\boldsymbol{P}_{\text{coarse}}$ is refined to yield $\boldsymbol{P}_{\text{fine}}$ using their surface normals and signed distances. For a detailed algorithm, please refer to Alg. 2.

In this algorithm, the background's boundary remains the boundary of the entire scene. For the foreground object, its boundary starts as the boundary of the entire scene and is expanded by $\delta = 0.1$ around its current surface points' boundary after each SP-MC iteration. The updated boundary is more accurate than the entire scene, leading to improved precision in SP-MC.

Finally, we emphasize the importance of the refinement step, which not only enhances the accuracy of $\boldsymbol{P}_{\text{coarse}}$, but also ensures reliable backpropagation of the gradient in the physical loss from an explicit physical simulator to the implicit SDF-based surface representation. Directly converting the SDF into a triangle mesh, as done in Marching Cubes [37], does not support stable and effective backpropagation in case of topological change, as observed in the works by Liao *et al.* [33] and Remelli *et al.* [56]. Thus, we detach the gradient in the steps involving the extraction of coarse surface points and rely on the refinement step for backpropagation.

### B.2   Comparison with Other Methods for Extracting Surface Points

To quantitatively compare with existing methods in efficiency, we assess SP-MC alongside Kaolin [13], which converts the SDF field into a triangle mesh and surface points. The assessment of time and memory involves the transformation from SDF to surface points, which are prepared

---

**Algorithm 2** Surface Points Marching Cubes (SP-MC) Algorithm

---

**Input:** SDF network $f(\cdot)$, Resolution $N$, Object Boundary $B$, Object Index $j$

**Output:** Surface Points $P_{\text{fine}}$

**function** SHIFT-INTERPOLATE($S$, $P$, *shift axis $v$*)

    // Grid shifting

    $S^{\text{shift}} \leftarrow S$ shift along $v$

    $M \leftarrow (S \circ S^{\text{shift}}) < 0$                       # $\circ$: *Hadamard product*, $M$: index mask

    $V \leftarrow P[M]$

    // Linear Interpolation

    $P_{\text{coarse}} = \left\{ p + \frac{S(p)}{S(p) - S(p^{\text{shift}})}(p^{\text{shift}} - p) \mid p \in V \right\}$      # $p^{\text{shift}}$: sign-flipping neighbor of $p$

    **return** $P_{\text{coarse}}$

**end function**

1: **Get $S \in \mathbb{R}^{N \times N \times N}$ and $P$ by Voxelization:**

2:     $P \leftarrow$ voxelize object boundary $B$ in resolution $N$

3:     $S \leftarrow \{ f_j(p) \mid p \in P \}$

4: **Get $P_{\text{coarse}}$ by Shift-Interpolate Operation:**

5:     $P_{\text{coarse}} \leftarrow \{ \text{SHIFT-INTERPOLATE}(S, P, v) \mid v \in \{x, y, z\} \}$

6: **Get $P_{\text{fine}}$ by Refinement Step:**

7:     $P_{\text{fine}} = \{ p - f(p) \cdot \nabla f(p) \mid p \in P_{\text{coarse}} \}$

8: **return** $P_{\text{fine}}$

---

and ready for use in both simulation and the computational graph for gradient backpropagation in both methods. We compared Kaolin and SP-MC at a grid resolution of 96 on a single A100 80GB machine, testing the average running time and GPU memory for all objects in all scenes of ScanNet++ [76]. From the results presented in Tab. A1, SP-MC consumes considerably less running time and GPU memory compared to Kaolin. The performance improvement of SP-MC primarily stems from its direct pursuit of surface points through simplified operations like grid-shifting and optimized parallel computation. This eliminates the need for face search in Kaolin's marching cubes, which consumes significant time and GPU memory.

Table A1: **Quantitative comparison between SP-MC and Kaolin.** SP-MC consumes less running time and GPU memory compared to Kaolin.

| | Running Time (s) ↓ | GPU Memory (MB) ↓ |
|---|---|---|
| Kaolin [13] | 0.482 | 21.327 |
| SP-MC | **0.264** | **13.673** |
| $\Delta$ | 0.218 (45.2%) | 7.654 (35.9%) |

Furthermore, we also note that Mezghanni *et al.* [43] propose a method for extracting surface points through direct thresholding of the SDF in the discretized signed distance field $S$, defined as:

$$P_{\text{coarse}} = \{ p \mid |s(p)| < \delta, p \in S \}. \tag{A16}$$

Although this method is conceptually simpler, it introduces a surface bias $\delta$, leading to an inevitable discrepancy between the extracted points and the actual surface points. It additionally requires a higher resolution of the SDF grid to capture fine structures, since the formulation puts higher requirements on the surface points, *i.e.*, from SDF sign flipping to SDF $< \delta$. Higher resolutions will decrease computational speed and consume more GPU memory.

## B.3   Supporting Plane

We use the supporting plane to provide contact and friction for object surface points in the physics simulation. More specifically, we use SP-MC to obtain the surface points of both the object to be simulated and the supporting plane (*e.g.*, background for objects on the floor) before each simulation. We only include the background surface points within proximity under the object surface points for the simulation to reduce the computational load. Note that our physical simulator and physical loss both support more general contacts, *e.g.*, the box on the cabinet and the monitor on the table

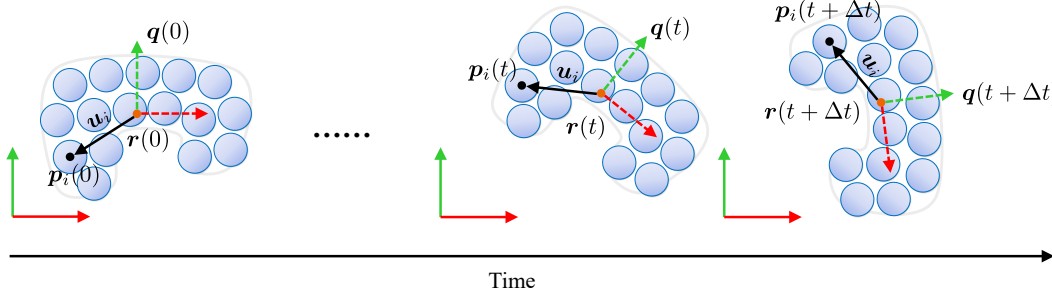

Figure A1: **Rigid body dynamics**. The world and rigid body coordinate system are represented by solid and dashed lines respectively.

demonstrated in Fig. 1. The only difference lies in the calculation of the supporting plane during simulation. For example, for the monitor on the table, we use the table surface points instead of background surface points to calculate the supporting plane.

## C   Particle-based Rigid Body Simulator

Our particle-based rigid body simulator is designed to enable convenient and efficient simulation of rigid bodies depicted as a collection of spherical particles of uniform size.

**Naming convention**   In this section, we represent vectors and second-order tensors with bold lowercase and uppercase letters, respectively; and denote scalars using italic letters. Superscripts are utilized to index rigid bodies, whereas subscripts are employed to index particles.

A rigid body simulator captures the body's dynamics through translation and rotation. The body's state is anchored to a reference coordinate system, which is initially aligned with the world coordinate system and located at the body's center of mass. The state at any given moment $t$ is encapsulated by the tuple:

$$\begin{pmatrix} \boldsymbol{r}(t) \\ \boldsymbol{q}(t) \\ \boldsymbol{v}(t) \\ \boldsymbol{\omega}(t) \end{pmatrix}, \tag{A17}$$

where $\boldsymbol{r}(t) \in \mathbb{R}^3$ and $\boldsymbol{q}(t) \in \mathbb{R}^4$ specify the position and orientation of the reference frame in relation to the world frame at time $t$, respectively. $\boldsymbol{q}(t) = [q_0, q_1, q_2, q_3]^T$ is a quaternion, where $q_0$ is a scalar value indicating the rotation angle, and $[q_1, q_2, q_3]$ is the normalized rotation axis. $\boldsymbol{v}(t) \in \mathbb{R}^3$ and $\boldsymbol{\omega}(t) \in \mathbb{R}^3$ represent the linear and angular velocities, respectively.

In our simulator, a rigid body is represented by a collection of spherical particles of uniform size, as depicted in Fig. A1. When a body consists of $N$ such particles, all with identical mass $m$ and initially position at $\boldsymbol{p}_i(0)$, the total mass of the system is:

$$M = N \cdot m. \tag{A18}$$

Furthermore, the system's center of mass, also serving as the origin of the reference frame $\boldsymbol{r}(0)$, is determined by:

$$\boldsymbol{r}(0) = \frac{\sum m \cdot \boldsymbol{p}_i(0)}{M}. \tag{A19}$$

Thus, the relative position of each particle within the local rigid body frame is $\boldsymbol{u}_i = [u_{i0}, u_{i1}, u_{i2}]^T = \boldsymbol{p}_i - \boldsymbol{r}(0)$. Consequently, the moment of inertia within the reference frame can be derived as:

$$\boldsymbol{I}_{\text{ref}} = \begin{bmatrix} \sum(u_{i1}{}^2 + u_{i2}{}^2)m & -\sum(u_{i0}u_{i1})m & -\sum(u_{i0}u_{i2})m \\ -\sum(u_{i0}u_{i1})m & \sum(u_{i0}{}^2 + u_{i2}{}^2)m & -\sum(u_{i1}u_{i2})m \\ -\sum(u_{i0}u_{i2})m & -\sum(u_{i1}u_{i2})m & \sum(u_{i0}{}^2 + u_{i1}{}^2)m \end{bmatrix}. \tag{A20}$$

The total mass $M$ and the moment of inertia $\boldsymbol{I}_{\text{ref}}$ are intrinsic physical properties of the rigid body, reflecting its resistance to external forces and torques. It is important to note that $\boldsymbol{I}$ is not constant as it varies with the body's orientation.

Table A2: **Three different contact status and their corresponding criteria.** Colliding Contact, Resting Contact and Separation

| Colliding Contact | Resting Contact | Separation |
|:---:|:---:|:---:|
| $\\\|\boldsymbol{p}_i^a - \boldsymbol{p}_j^b\\\| < 2r$ | $\\\|\boldsymbol{p}_i^a - \boldsymbol{p}_j^b\\\| < 2r$ | $\\\|\boldsymbol{p}_i^a - \boldsymbol{p}_j^b\\\| < 2r$ |
| $\boldsymbol{v}_c \cdot \boldsymbol{N}_c < -\epsilon$ | $\\\|\boldsymbol{v}_c \cdot \boldsymbol{N}_c\\\| < \epsilon$ | $\boldsymbol{v}_c \cdot \boldsymbol{N}_c > \epsilon$ |

## C.1 Forward Dynamics

According to Newton's second law, when subjected to external forces and moments, a rigid body's linear and angular velocities will change, altering its state. Using explicit Euler time integration, the evolution of a state variable between successive time steps can be summarized as follows:

$$
\begin{cases}
\boldsymbol{v}(t + \Delta t) = \boldsymbol{v}(t) + \Delta t M^{-1} \boldsymbol{f}(t) & \text{(A21a)} \\
\boldsymbol{r}(t + \Delta t) = \boldsymbol{r}(t) + \Delta t \boldsymbol{v}(t) & \text{(A21b)} \\
\boldsymbol{\omega}(t + \Delta t) = \boldsymbol{\omega}(t) + \Delta t \boldsymbol{I}^{-1}(t) \boldsymbol{\tau}(t) & \text{(A21c)} \\
\boldsymbol{q}(t + \Delta t) = \boldsymbol{q}(t) + [0, \dfrac{\Delta t}{2}] \times \boldsymbol{q}(t). & \text{(A21d)}
\end{cases}
$$

Here, $\Delta t$ is the time step size; $\boldsymbol{f} \in \mathbb{R}^3$ and $\boldsymbol{\tau} \in \mathbb{R}^3$ are the total external forces and torques, respectively. The inertia tensor of the rotated body $\boldsymbol{I}$ is given by:

$$
\boldsymbol{I}(t) = \boldsymbol{R}(t) \boldsymbol{I}_{\text{ref}} \boldsymbol{R}^T(t) \tag{A22}
$$

where the rotation matrix $\boldsymbol{R}(t)$ is derived from the corresponding quaternion $\boldsymbol{q}(t) = [q_0, q_1, q_2, q_3]^T$ using the formula:

$$
\boldsymbol{R}(t) = \begin{bmatrix}
2(q_0^2 + q_1^2) - 1 & 2(q_1 q_2 - q_0 q_3) & 2(q_1 q_3 + q_0 q_2) \\
2(q_1 q_2 + q_0 q_3) & 2(q_0^2 + q_2^2) - 1 & 2(q_2 q_3 - q_0 q_1) \\
2(q_1 q_3 - q_0 q_2) & 2(q_2 q_3 + q_0 q_1) & 2(q_0^2 + q_3^2) - 1
\end{bmatrix}. \tag{A23}
$$

During scene reconstruction, changes in the state of a rigid body are induced solely by gravity and contact. The gravity force is accounted for by setting $\boldsymbol{f} = M\boldsymbol{g}$, with $\boldsymbol{g}$ represents the acceleration of gravity. On the other hand, contact and friction forces are resolved using an impulse-based method, which is elaborated in the subsequent subsection.

## C.2 Collision Detection

In order to more realistically simulate the behavior of rigid bodies, we have to detect whether and where two rigid bodies come into contact with each other during dynamic motion. Since we represent rigid bodies as a set of particles, collision detection between complex-shaped rigid bodies can be simplified to relatively simple inter-particle collisions.

Specifically, for two particles that are sufficiently close (particle $i$ in rigid body $a$ and particle $j$ in rigid body $b$, both with a radius $r$), we can accurately approximate the contact position using the centers of the particles and define the contact normal as:

$$
\boldsymbol{N}_c = \frac{\boldsymbol{p}_i^a - \boldsymbol{p}_j^b}{\\\|\boldsymbol{p}_i^a - \boldsymbol{p}_j^b\\\|}, \tag{A24}
$$

and the relative velocity at the contact points is given by:

$$
\boldsymbol{v}_c = (\boldsymbol{v}^a + \boldsymbol{R}^a \boldsymbol{u}_i^a) - (\boldsymbol{v}^b + \boldsymbol{R}^b \boldsymbol{u}_j^b), \tag{A25}
$$

with the normal and tangential component are defined as:

$$
\begin{cases}
\boldsymbol{v}_{c\perp} = (\boldsymbol{v}_c \cdot \boldsymbol{N}_c) \boldsymbol{N}_c & \text{(A26a)} \\
\boldsymbol{v}_{c\\\|} = \boldsymbol{v}_c - \boldsymbol{v}_{c\perp}. & \text{(A26b)}
\end{cases}
$$

Their relative contact status can be categorized into three types based on the criteria illustrate in Tab. A2. Among these, only colliding and resting contact require further processing in the simulation pipeline.

## C.3 Colliding Contact

The criterion $\boldsymbol{v}_c \cdot \boldsymbol{N}_c < -\epsilon$ indicates that the two particles are approaching each other and further interpenetration will occur, therefore the simulator must separate them. The fundamental principle of impulse-based rigid body contact simulation [67] lies in instantaneously adjusting the velocity to prevent subsequent interpenetration. This method eliminates the need for directly applying force over an extended period.

Adhering to the principle of conservation of momentum and Coulomb's friction model, we know the physically plausible relative velocity $\boldsymbol{v}_c^* = \boldsymbol{v}_{c\perp}^* + \boldsymbol{v}_{c\|}^*$ after contact should be:

$$
\begin{cases}
\boldsymbol{v}_{c\perp}^* = -\mu \boldsymbol{v}_{c\perp} & \text{(A27a)} \\
\boldsymbol{v}_{c\|}^* = \alpha \boldsymbol{v}_{c\|} & \text{(A27b)} \\
\alpha = \max(1 - \dfrac{\eta(1+\mu)\|\boldsymbol{v}_{c\perp}\|}{\|\boldsymbol{v}_{c\|}\|}, 0), & \text{(A27c)}
\end{cases}
$$

where $\mu$ is the coefficient of restitution and $\eta$ is the friction coefficient. To achieve the desired velocity $\boldsymbol{v}_c^*$, the required impulse $\boldsymbol{J}$ at the contact point is calculated as:

$$
\begin{cases}
\boldsymbol{J} = \boldsymbol{K}^{-1}(\boldsymbol{v}_c^* - \boldsymbol{v}_c) & \text{(A28a)} \\
\boldsymbol{K} = \dfrac{\mathbb{I}}{M^a} + \dfrac{\mathbb{I}}{M^b} - [\boldsymbol{R}^a\boldsymbol{u}_i^a]_\times(\boldsymbol{I}^a)^{-1}[\boldsymbol{R}^a\boldsymbol{u}_i^a]_\times - [\boldsymbol{R}^b\boldsymbol{u}_j^b]_\times(\boldsymbol{I}^b)^{-1}[\boldsymbol{R}^b\boldsymbol{u}_j^b]_\times, & \text{(A28b)}
\end{cases}
$$

where $\mathbb{I}$ represents the $3 \times 3$ identity matrix and the operator $[\ \ ]_\times$ transform a vector into a skew-symmetric matrix. Under the influence of $J$, the linear and angular velocity of the involved rigid bodies are updated as follows:

$$
\begin{cases}
\boldsymbol{v}^a = \boldsymbol{v}^a + \dfrac{1}{M^a}\boldsymbol{J} & \text{(A29a)} \\
\boldsymbol{\omega}^a = \boldsymbol{\omega}^a + (\boldsymbol{I}^a)^{-1}(\boldsymbol{R}^a\boldsymbol{u}_i^a \times \boldsymbol{J}) & \text{(A29b)} \\
\boldsymbol{v}^b = \boldsymbol{v}^b - \dfrac{1}{M^b}\boldsymbol{J} & \text{(A29c)} \\
\boldsymbol{\omega}^b = \boldsymbol{\omega}^b - (\boldsymbol{I}^b)^{-1}(\boldsymbol{R}^b\boldsymbol{u}_j^b \times \boldsymbol{J}). & \text{(A29d)}
\end{cases}
$$

These updates ensure that the bodies respond correctly to the collision, separating or bouncing off each other in a manner that conserves momentum and energy as dictated by the specified restitution coefficient.

When multiple particles on a rigid body collide simultaneously, we use the average linear and angular impulses to update the rigid body's linear and angular velocities. If one of the rigid body during in contact is a fixed boundary, such as the floor, simply setting its corresponding $1/M$ and $\boldsymbol{I}^{-1}$ to zero will adequately handle the situation.

## C.4 Resting Contact

The criterion $\|\boldsymbol{v}_c \cdot \boldsymbol{N}_c\| < \epsilon$ indicates that an object maintains continuous contact with another without significant changes in position or orientation, such as a chair resting on the floor. Achieving stable resting contact without objects slowly penetrating each other or jittering due to numerical errors can be challenging with physics-based method.

To simplify this, our simulator adopt a strategy commonly employed in game development and robotics to enhance performance and realism. If a dynamic rigid body remains stationary or moves extremely slowly for a few seconds, our simulator will mark it as sleeping. Once classified as sleeping, the rigid body will be temporarily excluded from all steps in the simulation pipeline, except for collision detection. When the sleeping body come into colliding contact with another non-sleeping rigid body, it will automatically "wake up" and get back into the simulation again.

In practical implementation, we employ the following method to quantitatively assess the motion of a rigid body over a short historical period [46]:

$$
\text{rwa} = \gamma \cdot \text{rwa} + (1 - \gamma) \cdot v_{\text{up}}^2, \tag{A30}
$$

where the weight $\gamma$ lies within the range $[0, 1]$; $v_{\mathrm{up}}$ represents the upper bound of the current speed of the rigid body [46], which can be effectively approximated by:

$$v_{\mathrm{up}}^2 = (\boldsymbol{v} + \boldsymbol{R}\boldsymbol{u}_{\max} \times \boldsymbol{\omega})^T (\boldsymbol{v} + \boldsymbol{R}\boldsymbol{u}_{\max} \times \boldsymbol{\omega}) \tag{A31}$$

$$\approx 2 \cdot (\boldsymbol{v}^T\boldsymbol{v} + (\boldsymbol{\omega}^T\boldsymbol{\omega}) \cdot (\boldsymbol{u}_{\max}^T\boldsymbol{u}_{\max})). \tag{A32}$$

Here $\boldsymbol{u}_{\max}$ denotes the maximum distance from any particle on the surface to the body's center of mass, calculated once during the initialization phase. Given that our simulator is solely affected by gravity, a body is set to sleep if rwa meets the following condition:

$$\mathrm{rwa} < \|\boldsymbol{g}\| \cdot \Delta t. \tag{A33}$$

### C.5 Implementation Details

For the sake of reproducible, we present the pipeline of our particle-based rigid body simulation during each time step as in Alg. 3. Our simulator and its gradient support are developed using DiffTaichi [19], which is a high-performance differentiable physical programming language designed for physical simulations, and computational science.

For all of our examples, we set time step $\Delta t = 0.01\,\mathrm{s}$; particle radius $r = 0.005\,\mathrm{m}$; particle mass $m = 0.01\,\mathrm{kg}$; the coefficient of restitution $\mu = 0.0$, the friction coefficient $\eta = 0.4$, the relative velocity criterion $\epsilon = 1e - 5$ and $\gamma = 0.1$.

---

**Algorithm 3** The pipeline of our particle-based rigid body simulator

---

1: **Input:** Initial particle positions $\boldsymbol{p}^i(0)$ for each rigid body $i$ in the scene.
2: **Output:** Final particle positions $\boldsymbol{p}^i(t)$ when their belonging rigid body reaches stable equilibrium, with flags for all particles that have collided.
3:
4: // physical properties
5: **For** *each rigid body* **do**
6:     compute mass and center of mass: $M \leftarrow$ Eq. (A18), $\boldsymbol{r}(0) \leftarrow$ Eq. (A19)
7:     **For** *each particle* **do**
8:         compute particle position in reference frame: $\boldsymbol{u}_i \leftarrow \boldsymbol{p}_i(0) - \boldsymbol{r}(0)$
9:     **end for**
10:     compute inertia matrix in reference frame: $\boldsymbol{I}_{\mathrm{ref}} \leftarrow$ Eq. (A20)
11:     compute the maximum distance: $u_{\max} \leftarrow \max(\boldsymbol{u}_i)$
12: **end for**
13:
14: **For** *time step $t$* **do**
15:     // forward dynamics
16:     **For** *each awake rigid body* **do**
17:         apply gravity force: $\boldsymbol{f}(t) \leftarrow M\boldsymbol{g}$
18:         compute linear and angular velocities $\boldsymbol{v}(t) \leftarrow$ Eq. (A21a), $\boldsymbol{\omega}(t) \leftarrow$ Eq. (A21b)
19:         compute position and orientation: $\boldsymbol{r}(t) \leftarrow$ Eq. (A21c), $\boldsymbol{q}(t) \leftarrow$ Eq. (A21d)
20:         compute rotation matrix: $\boldsymbol{R}(t) \leftarrow$ Eq. (A23)
21:     **end for**
22:
23:     // update particles
24:     **For** *each particle* **do**
25:         update particle position in world space: $\boldsymbol{p}_i(t) \leftarrow \boldsymbol{R}(t)\boldsymbol{u}_i + \boldsymbol{r}(t)$
26:     **end for**
27:
28:     // collision detection and colliding contact resolve
29:     **While** *no colliding contact* **do**
30:         **For** *each pair of particles* **do**
31:             **if** $\|\boldsymbol{p}_i^a - \boldsymbol{p}_j^b\| < 2r$ **then**
32:                 compute contact normal: $\boldsymbol{N}_c \leftarrow Eq.$ (A24)
33:                 compute contact velocity: $\boldsymbol{v}_c \leftarrow Eq.$ (A25), $\boldsymbol{v}_{c\perp} \leftarrow Eq.$ (A26a)
34:                 **if** $\boldsymbol{v}_c \cdot \boldsymbol{N}_c < \epsilon$ **then**
35:                     compute desired contact velocity: $\boldsymbol{v}_c^* \leftarrow Eq.$ (A27a), $Eq.$ (A27a)

```
36:                    compute required impulse: J ← Eq. (A28a)
37:                        if ‖v_{c⊥}‖ > r/Δt
38:                            reactive the rigid body
39:                        end if
40:                    end if
41:                end if
42:            end for
43:        end while
44:
45:        // update rigid body state
46:        For each rigid body do
47:            update linear and angular velocities: v(t) ← Eq. (A29a), ω(t) ← Eq. (A29a)
48:        end for
49:
50:        // resting contact
51:        For each rigid body do
52:            compute historical motion information v_{up} ← Eq. (A32), rwa ← Eq. (A30)
53:                if rwa < ‖g‖ · Δt then
54:                    set the body to sleep
55:                    v ← [0, 0, 0], ω ← [0, 0, 0]
56:                end if
57:        end for
58: end for
```

# D    More Experiment Details

## D.1    Data Preparation

**Monocular Cues**    We utilize a pre-trained Marigold model [27] to generate a depth map $\bar{D}$ for each input RGB image. It's important to note that estimating the absolute scale in general scenes is challenging, thus $\bar{D}$ should be regarded as a relative depth cue. Furthermore, we employ another pre-trained Omnidata model [12] to obtain normal maps $\bar{N}$ for each RGB image. While depth cues offer semi-local relative information, normal cues are local and capture geometric intricacies. Consequently, we anticipate that surface normals and depth complement each other effectively.

**GT Instance Mask**    For the ScanNet++ [76] dataset, we utilize the provided GT mesh and per-vertex 3D instance annotations, along with their rendering engine to generate instance masks for each image. For the ScanNet [9] and Relica [62] datasets, we observed discrepancies in the masks provided by RICO [32] and ObjectSDF++ [69]. To ensure a fair comparison with the baselines, we merged their instance masks into consistent ones. In our experiments, we focused solely on object-ground support for simplicity and training efficiency, leaving the determination of more general support relationships for future work.

## D.2    Evaluation Metrics

**Stability Ratio**    To evaluate the physical stability of a reconstructed object mesh shape, we employ the Isaac Gym [39] simulator. This involves conducting a dropping simulation to determine if the shape remains stable within $5°$ in rotation and $5cm$ translation after the simulation, under the influence of gravity, contact forces, and friction provided by the ground. Next, we define the stability ratio of the scene as the proportion of the number of stable objects to the total object number in the scene. More specifically, we import the object shape and the reconstructed background into the simulator using URDFs that include parameters such as the center of mass, mass, and inertia matrix, where the relative positions of the object shape and background are preserved in the scene. Subsequently, we simulate for $T = 200$ steps with a time step of $\Delta t = 0.016s$ (i.e. $60Hz$).

**Reconstruction Metrics**    To evaluate 3D scene and object reconstruction, we use the CD in $cm$, F-score with a threshold of $5cm$ and NC following prior research [77, 32, 69]. In detail, CD comes from *Accuracy* and *Completeness*, F-score is derived from *Precision* and *Recall*, and NC is computed using both *Normal-Accuracy* and *Normal-Completeness*. We follow previous work [18, 77, 32, 69] to evaluate reconstruction only on the visible areas. These metrics are defined in Tab. A3.

Table A3: **Evaluation metrics.** We show the evaluation metrics with their definitions that we use to measure reconstruction quality. $P$ and $P^*$ are the point clouds sampled from the predicted and the ground truth mesh. $\boldsymbol{n_p}$ is the normal vector at point $\boldsymbol{p}$.

| Metric | Definition |
|---|---|
| **Chamfer Distance (CD)** | $\frac{Accuracy+\text{Completeness}}{2}$ |
| *Accuracy* | $\operatorname*{mean}_{\boldsymbol{p}\in P}\left(\min_{\boldsymbol{p^*}\in P^*}\lVert\boldsymbol{p}-\boldsymbol{p^*}\rVert_1\right)$ |
| *Completeness* | $\operatorname*{mean}_{\boldsymbol{p^*}\in P^*}\left(\min_{\boldsymbol{p}\in P}\lVert\boldsymbol{p}-\boldsymbol{p^*}\rVert_1\right)$ |
| **F-score** | $\frac{2\times\text{Precision}\times\text{Recall}}{\text{Precision}+\text{Recall}}$ |
| *Precision* | $\operatorname*{mean}_{\boldsymbol{p}\in P}\left(\min_{\boldsymbol{p^*}\in P^*}\lVert\boldsymbol{p}-\boldsymbol{p^*}\rVert_1 < 0.05\right)$ |
| *Recall* | $\operatorname*{mean}_{\boldsymbol{p^*}\in P^*}\left(\min_{\boldsymbol{p}\in P}\lVert\boldsymbol{p}-\boldsymbol{p^*}\rVert_1 < 0.05\right)$ |
| **Normal Consistency** | $\frac{Normal\ Accuracy+Normal\ Completeness}{2}$ |
| *Normal Accuracy* | $\operatorname*{mean}_{\boldsymbol{p}\in P}\left(\boldsymbol{n_p}^T\boldsymbol{n_{p^*}}\right)\ \text{s.t.}\ \boldsymbol{p^*}=\arg\min_{\boldsymbol{p^*}\in P^*}\lVert\boldsymbol{p}-\boldsymbol{p^*}\rVert_1$ |
| *Normal Completeness* | $\operatorname*{mean}_{\boldsymbol{p^*}\in P^*}\left(\boldsymbol{n_p}^T\boldsymbol{n_{p^*}}\right)\ \text{s.t.}\ \boldsymbol{p}=\arg\min_{\boldsymbol{p}\in P}\lVert\boldsymbol{p}-\boldsymbol{p^*}\rVert_1$ |

### D.3  Intermediate Results

In Fig. A2, we present the reconstruction results of the intermediate states across different stages. Epochs 360 and 430 mark the beginning of the 2$^{\text{nd}}$ and 3$^{\text{rd}}$ training stages. In the 1$^{\text{st}}$ stage, the reconstruction quality of the object's visible structure is continuously improved. In the 2$^{\text{nd}}$ stage, the object areas crucial for stability are optimized with the physical uncertainty. In the 3$^{\text{rd}}$ stage, the reconstructed object reaches stability when the physical loss is introduced.

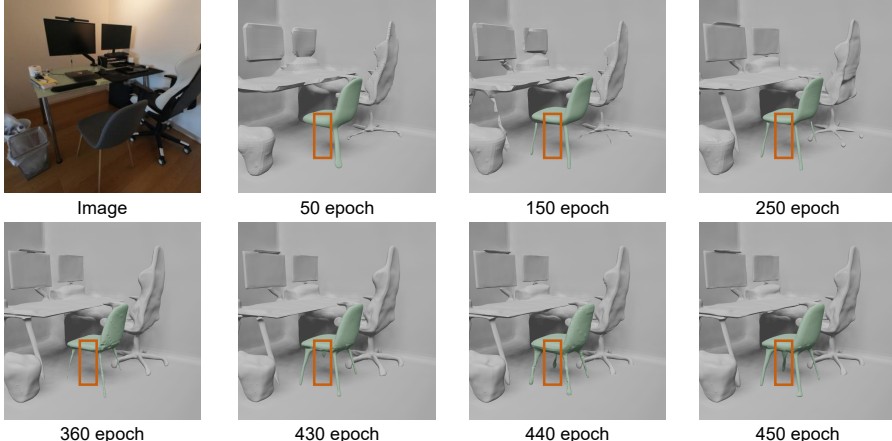

Figure A2: **Visualization of intermediate results during training.** Epochs 360 and 430 mark the beginning of the 2$^{\text{nd}}$ and 3$^{\text{rd}}$ training stages.

## D.4 Time Consumption

We conduct additional experiments on example scenes from the ScanNet++ dataset to assess the time consumption of the rendering and physical simulation. On average, it takes approximately 0.23 seconds (5.13% in total time consumption) for rendering and 4.25 seconds (94.87% in total time consumption) for physical simulation (including automatic gradient calculation) during one single forward pass. Note that the time required for physical simulation is related to the number of particle points involved, it will vary across different scenes and objects. The above experiments involve around 8000 object surface points and 6000 background surface points (i.e. the supporting plane) on average physical simulation.

Additionally, we compare the total training time between our method, RICO, and ObjectSDF++ as shown in Tab. A4 on ScanNet++. Integrating physical simulation has significantly enhanced the stability of the reconstruction results, although it has also increased time consumption.

Table A4: **Training time comparison.**

|  | RICO | ObjectSDF++ | Ours |
|---|---|---|---|
| Stage 1 (0-360 epoch) | 6.86h | 7.14h | 6.91h |
| Stage 2 (360-430 epoch) | 1.45h | 1.43h | 2.46h |
| Stage 3 (430-450 epoch) | 0.39h | 0.41h | 3.19h |
| Total | 8.70h | 8.98h | 12.56h |

## D.5 Performance Sensitivity Analysis

**Performance vs. Max Simulation Steps in a Forward Simulation**  In our framework, the physics forward simulation continues until the object reaches a stable state or the maximum simulation steps are reached. Therefore, the choice of maximum simulation steps affects both simulation time and performance. We conducted a sensitivity analysis of performance versus maximum simulation steps in a forward simulation. The results are evaluated on example scenes in ScanNet++, shown in Tab. A5. The results illustrate that increasing the maximum simulation steps generally improves the final stability of the objects. This trend becomes negligible once the maximum simulation steps exceed 100, as most objects achieve a stable state within 100 simulation steps. This also explains why increasing the maximum simulation steps leads to longer simulation time, though this increase is not linear to the number of steps. Consequently, in the experiments for our main paper, we chose 100 as the maximum simulation steps.

Table A5: **Performance of varying maximum simulation steps in one forward simulation.**

| Max Sim. Steps | CD ↓ scene/obj | F-Score ↑ scene/obj | NC ↑ scene/obj | SR (%) ↑ | Total time | Stage-3 time |
|---|---|---|---|---|---|---|
| 25 | 2.91/3.24 | 88.17/86.61 | 91.56/85.04 | 69.23 | 10.64h | 1.27h |
| 50 | 2.83/3.26 | 88.41/86.66 | 91.82/85.04 | 69.23 | 11.45h | 2.08h |
| 75 | **2.78/3.16** | **88.84/87.12** | **91.84/85.68** | 76.92 | 12.08h | 2.71h |
| 100 | 2.86/3.24 | 88.34/86.58 | 91.72/85.12 | **84.62** | 12.56h | 3.19h |
| 125 | 2.88/3.28 | 88.19/86.48 | 91.54/85.08 | **84.62** | 12.92h | 3.55h |
| 150 | 2.91/3.25 | 88.24/86.52 | 91.53/85.06 | **84.62** | 13.26h | 3.89h |

**Performance vs. Total Simulation Epochs.**  We conducted a sensitivity analysis of performance versus total simulation epochs to discuss the relationship between training time and performance after adding the physical loss in stage 3. The results are shown in Tab. A6, tested on ScanNet++. The table shows that as the number of training epochs with physical loss increases, the stability of the objects improves. However, there is no significant improvement after 445 epochs, as most objects had already achieved stability. Notably, the longest training time was observed between 430 and 435 epochs, after which the time required for simulation progressively decreased, again due to the improved object stability.

Table A6: **Performance of increasing the number of training epochs with physical simulations.**

| Training epoch | CD ↓ scene/obj | F-Score ↑ scene/obj | NC ↑ scene/obj | SR (%) ↑ | Total time | Δ time |
|---|---|---|---|---|---|---|
| 430 (no sim.) | 2.95/3.31 | 87.84/86.32 | 90.42/84.83 | 61.54 | 9.37h | 0 |
| 435 | 2.88/3.31 | 88.06/86.35 | 90.93/84.36 | 69.23 | 10.49h | 1.12h |
| 440 | **2.81/3.17** | **88.75/87.03** | **91.93/85.77** | 76.92 | 11.39h | 0.90h |
| 445 | 2.82/3.22 | 88.54/86.96 | 91.83/85.41 | **84.62** | 12.10h | 0.71h |
| 450 | 2.86/3.24 | 88.34/86.58 | 91.72/85.12 | **84.62** | 12.56h | 0.46h |

# E    Limitation

Representative failure examples in Fig. 8 present that optimizing with the physical loss may lead to degenerated object shapes, *e.g.*, bulges, to maintain physical stability. This is due to insufficient supervision from the rendering losses. The disconnected parts in the reconstruction results are another common limitation of the current neural implicit surface representation and reconstruction pipeline. This may be addressed by incorporating topological regularization to penalize disconnected parts of the object or further enhancing the simulator. Plus, our particle-based simulator treats all surface points from an object as a rigid body, thus it cannot handle soft bodies (which deform after collision) or dynamic scenes.

In addition, our current framework requires additional object-supporting information for the simulation. While determining ground-object support is straightforward, identifying more complex relationships remains challenging. The implicit representation is optimized through per-object physical simulation with the background, for the sake of efficient computation in the current neural scene understanding settings. However, our SP-MC, physical simulator and loss are designed to be compatible with multi-object scenarios, enabling seamless extension to joint optimization for the whole scene, ensuring versatility without loss of generality.

# F    Potential Negative Impact

3D scene reconstruction in general, while offering various benefits in fields like AR/VR, robotic and Embodied AI, also raises concerns about potential negative social impacts. Some of these impacts include potential privacy concerns in public areas, surveillance and security risks. Addressing these concerns requires careful consideration of ethical guidelines, regulatory frameworks, and responsible development practices to ensure that 3D scene reconstruction is deployed in a manner that respects privacy, security, and societal well-being.

