# OpenReview forum: "PhyRecon: Physically Plausible Neural Scene Reconstruction"
_NeurIPS.cc/2024/Conference — NeurIPS 2024 poster_

### Official Review · Reviewer_qV3n · 2024-07-07

**Soundness:** 4
**Presentation:** 4
**Contribution:** 3
**Rating:** 7
**Confidence:** 4

**Summary:**

This paper addresses scene surface reconstruction in an object-compositional format. Unlike previous work that only considered rendering constraints, the proposed PhyRecon framework incorporates physical constraints into the model training. To identify interaction points, the paper introduces a differentiable transformation between the SDF field and explicit surface points. Using these points, the authors integrate physical constraints into previous approaches to ensure reasonable interactions between objects. To further enhance model performance, they introduce a joint uncertainty modeling design for depth, normal, and physical constraints, which increases the stability of the final reconstruction results. The experimental results demonstrate the effectiveness of the framework and its components, proving beneficial for various downstream tasks.

**Strengths:**

1. The overall framework is mind-blowing and insightful. Extending the neural implicit representation, which is mainly supervised by rendering loss, to include physical constraints is non-trivial. This work proposes an innovative solution in this direction with well-reasoned designs, making it more applicable to real-world applications like robotics and virtual reality.
2. The physical constraint is novel and enhances the fidelity of object reconstruction within a compositional design. The proposed framework offers a comprehensive solution with several interesting components, such as SP-MC and physical-guided pixel sampling.
3. The experimental results are extensive, showing significant improvements in object reconstruction quality and stability compared to previous baselines.
4. The physical pixel-guided sampling is an interesting and important trick that enhances the effectiveness of the physical constraint.

**Weaknesses:**

1. The overall solution is based on ObjectSDF++ and RICO, which requires hours of optimization for training. The introduction of additional constraints in PhyRecon further increases the total training time. Improving efficiency could be a focus for future work.
2. The training process involves three stages: initially applying rendering loss for initialization, then incorporating a physical simulator, and finally introducing a physical loss to stabilize object contact. This multi-stage optimization and the complexity of the loss functions make the entire system intricate.

**Questions:**

1. The method relies on ground-truth instance labels. It would be interesting to explore whether the proposed physical solution can handle noise in the ground-truth 2D instance mask, such as erosion and dilation. While uncertainty training might help mitigate this issue, it is crucial to understand how much the physical constraint can aid in overcoming label noise.
2. In the particle simulator, how does the system perform when the points are sparse and non-uniformly distributed? Understanding the simulator's effectiveness in such cases would be beneficial.

**Limitations:**

The author mentioned their limitation and potential societal impact in the appendix.

---

> ### Author Rebuttal · Authors · 2024-08-07
>
> We sincerely thank the reviewer for the constructive comments on our work. We will thoroughly address each of the concerns raised. Notably, we have added visualizations of intermediate results during training in the global rebuttal PDF. We hope these visualizations will help address your questions about the multi-stage optimization.
>
> **1. Comparison of total training time between our method, RICO, and ObjectSDF++.**
>
> We compare the total training time between our method, RICO, and ObjectSDF++ as shown in the following table (test on example scenes from ScanNet++):
>
> |                         | RICO  | ObjectSDF++ | Ours  |
> | ---------------------   | ----- | ----------- | ----- |
> | Stage 1 (0-360 epoch)   | 6.86h |    7.14h    | 6.91h |
> | Stage 2 (360-430 epoch) | 1.45h |    1.43h    | 2.46h |
> | Stage 3 (430-450 epoch) | 0.39h |    0.41h    | 3.19h |
> | total                   | 8.70h |    8.98h    | 12.56h |
>
> Stage 1: We use rendering uncertainty to adjust the rendering loss.
>
> Stage 2: We add physical uncertainty to adjust the rendering loss and guide pixel sampling.
>
> Stage 3: We incorporate the physical loss.
>
> Integrating physical simulation has significantly enhanced the stability of the reconstruction results, although it has also increased time consumption. Exploring 3DGS-based methods [1, 2, 3] to improve scene reconstruction speed and further accelerating simulation computations will be interesting research directions.
>
> **2. This multi-stage optimization and the complexity of the loss functions make the entire system intricate.**
>
> In our paper, we use a three-stage training method for two main reasons:
> * Simulation requires a reasonably initialized shape to ensure correct contact points.
> * Compared to rendering calculations, adding simulation increases the computational cost, so we minimize the use of physical simulation as much as possible.
>
> Therefore, we initially use only rendering loss to optimize and achieve a reasonable initial shape in stage 1. We then introduce physical simulation to incorporate physical uncertainty in stage 2. In this stage, each object undergoes simulation once at the beginning of each epoch, further optimizing the object's shape, particularly in regions crucial for physical stability. In stage 3, we include physical loss, and physical simulation is applied at each iteration of an epoch. This stage has the highest computational load due to the added physical simulation at each iteration, but it is crucial for enhancing object stability. Conversely, introducing physical simulation and physical loss from the start would not only excessively increase the computational load but also hinder shape optimization due to incorrect collision information.
>
> Additionally, in the global rebuttal Fig.1, we present the reconstruction results of the intermediate states across different stages:
>
> Stage 1: as shown at 50, 150, 250, 360 epochs.
>
> Stage 2: as shown at 360, 430 epochs.
>
> Stage 3: as shown at 430, 440, 450 epochs.
>
> Additionally, all loss functions in our method plays an important role in optimization. Furthermore, in practical implementations, thanks to our adjustment flexibility of rendering uncertainty and physical uncertainty, the various loss weights do not require meticulous tuning for each scene to achieve good results.
>
> In summary, although the multi-stage training and loss designs may appear complex, they exhibit flexibility in deployment and robustness across various scenes. Future research could focus on reducing system complexity while preserving current performance levels.
>
> **3. Whether the proposed physical solution can handle noise in the ground-truth 2D instance mask?**
>
> Yes, the physical solution we proposed, including physical uncertainty and physical loss, can alleviate noise in the ground-truth 2D instance mask. For example, in Fig.6 of the main paper, the final reconstruction result of our method includes legs which the GT instance mask of the chair lacks. Moreover, with the rapid development of segmentation and tracking models, e.g., SAM [4] and SAM2 [5], future efforts can be devoted to decompositional neural scene reconstruction with object masks from images obtained with off-the-shelf models.
>
> **4. How does the system perform when the points are sparse and non-uniformly distributed?**
>
> Based on our current implementations, sparsely and non-uniformly distributed particles may lead to the following two problems in contact resolution:
> * Missed collision detections: objects may pass through the ground.
> * Numerical instability: too sparse points will affect the calculation of impulse J (eq.A27a and eq.A27b) in supp Sec.C.3 Colliding Contact.
>
> Thus, in our paper, we propose the SP-MC with an adaptive object bounding box (as discussed in lines 912-915) to effectively address these issues by ensuring a uniform and dense surface particle distribution.
> Additionally, extending our particle-based rigid body simulator to accommodate particles of varying sizes and anisotropy would be an interesting future direction, thereby better supporting low-quality particle distributions.
>
> [1] 3D Gaussian Splatting for Real-Time Radiance Field Rendering
>
> [2] 2D Gaussian Splatting for Geometrically Accurate Radiance Fields
>
> [3] High-quality Surface Reconstruction using Gaussian Surfels
>
> [4] Segment and Track Anything
>
> [5] SAM 2: Segment Anything in Images and Videos

---

### Official Review · Reviewer_fqR5 · 2024-07-08

**Soundness:** 4
**Presentation:** 3
**Contribution:** 3
**Rating:** 7
**Confidence:** 5

**Summary:**

The paper introduces a novel method for physically plausible 3D scene reconstruction. It enforces the reconstructions to be in a static equilibrium state, which is achieved by minimizing the displacement of 3D points over time in a differentiable rigid body simulator. The experiments are conducted on scannet/replica/scannet++, showing quite significant improvement over prior methods.

**Strengths:**

- The paper is easy to follow, and the visual comparisons are adequately made.
- There are good technical contributions and the method is sound.
  - They present a differentiable pipeline that can propagate gradients from after-simulation states to the reconstruction, which is achieved by a differentiable mapping from neural SDF to particles (SP-MC), and explicit Euler integration for simulation (which is already differentiable).
  - They also develop a re-weighting strategy that down-weights the loss terms with large uncertainty. The choice of uncertainty is different from term to term, measurement uncertainty is used fo depth and normal; physical uncertainty (as defined by the displacement of point over simulation) is used for segmentation as a proxy. The use of uncertainty has two major benefits: (1) the noise in the depth/normal/segmentation priors are filtered out and (2) the pixel sampling can focus on regions with high uncertainty to enhance the reconstruction quality.
- The results are strong. The comparisons clearly shows the improvement over prior methods. They also compared to prior works quantitatively, showing better performance in terms of reconstruction quality, normal consistency, and object stability.
- The authors ablated the design choices of physical loss, uncertainty and sampling strategy, showing the benefit of adding each component.

**Weaknesses:**

- It only deals with static scenes with rigid objects. The physical loss presented is not applicable to complex contact interactions, dynamic scenes or soft materials. But I don't think this is a major issue as the method works well for the indoor scene reconstruction task being considered.
- More explanation on SP-MC is needed. Since the gradient only comes from the refinement step, I would imagine the gradient to be fairly local and noisy. How does it work when the initial shape is far from being physically plausible?
- I'd like to suggest moving appendix C to the main text, since the physical simulator plays an important part in the overall story.

**Questions:**

- How expensive is the simulation compared to rendering? e.g., in a forward pass, how much time is consumed by simulation? Does the use of simulation introduce significant compute overhead? Relatedly, I'd like to see a sensitivity analysis of performance vs number of simulation steps.
- How is the ground-plane computed for physical simulation? Does the method handle complex terrian (e.g., with a carpet on the floor, or outdoor scenes)?
- Since the optimization is multi-staged, it will be good to show result after each stage.
- Showing more intermediate results along the axis of optimization would help. For instance, how does the shape change from iteration 0 to final iteration?

**Limitations:**

The paper discussed the limitations and social impact.

---

> ### Author Rebuttal · Authors · 2024-08-07
>
> We sincerely thank the reviewer for the constructive comments on our work. We will thoroughly address each of the concerns raised. Notably, we have added visualizations of intermediate results during training in the global rebuttal PDF, along with sensitivity experiments regarding the max simulation steps in a forward simulation and the total simulation epochs. We hope these additions help clarify your questions.
>
> **1. More intermediate results during optimization, particularly after each stage.**
>
> In the global rebuttal Fig.1, we present the reconstruction results of the intermediate states across different stages:
>
> Stage 1: We use rendering uncertainty to adjust the rendering loss, as shown at 50, 150, 250, and 360 epochs.
>
> Stage 2: We add physical uncertainty to adjust the rendering loss and guide pixel sampling, as shown at 360 and 430 epochs.
>
> Stage 3: We incorporate the physical loss, as shown at 430, 440, and 450 epochs.
>
> In the Stage 1, the quality of the object's thin structure reconstruction is continuously improved. In the Stage 2, critical parts contributing to the object's stability are significantly optimized. In the Stage 3, the stability of the object reconstruction is greatly enhanced.
>
> **2. Comparison of time consumption between physical simulation and rendering.**
>
> We conduct additional experiments on example scenes from ScanNet++ dataset to assess the time consumption of the rendering and physical simulation. On average, it takes approximately 0.23 seconds (5.13% in total time consumption) for rendering and 4.25 seconds (94.87% in total time consumption) for physical simulation (including automatic gradient calculation) during one single forward pass. Note that the time required for physical simulation is related to the number of particle points involved, it will vary across different scenes and objects.  The above experiments involves around 8000 object surface points and 6000 background surface points (i.e. the supporting plane) on average physical simulation.
>
> Additionally, we compare the total training time between our method, RICO, and ObjectSDF++ as shown in the following table on ScanNet++:
>
> |                         | RICO  | ObjectSDF++ | Ours  |
> | ---------------------   | ----- | ----------- | ----- |
> | Stage 1 (0-360 epoch)   | 6.86h |    7.14h    | 6.91h |
> | Stage 2 (360-430 epoch) | 1.45h |    1.43h    | 2.46h |
> | Stage 3 (430-450 epoch) | 0.39h |    0.41h    | 3.19h |
> | total                   | 8.70h |    8.98h    | 12.56h |
>
> Integrating physical simulation has significantly enhanced the stability of the reconstruction results, although it has also increased time consumption. Further accelerating simulation computations will be an interesting research direction. We will add these additional results and discussions in the revision.
>
> **3. Sensitivity analysis of performance vs max simulation steps in a forward simulation.**
>
> In our framework, the physics forward simulation continues until the object reaches a stable state or the maximum simulation steps are reached. Therefore, the choice of maximum simulation steps affects both simulation time and performance. We conducted a sensitivity analysis of performance versus maximum simulation steps in a forward simulation. The results are evaluated example scenes on ScanNet++, shown in the global rebuttal Tab.1.
>
> The table illustrates that increasing the maximum simulation steps generally improves the final stability of the objects. This trend becomes negligible once the maximum simulation steps exceed 100, as most objects achieve a stable state within 100 simulation steps. This also explains why increasing the maximum simulation steps leads to longer simulation times, though this increase is not linear with respect to the number of steps. Consequently, in the experiments for our main paper, we chose 100 as the maximum simulation steps.
>
> **4. Sensitivity analysis of performance vs total simulation epochs.**
>
> To discuss the relationship between training time and performance after adding the physical loss in stage 3, we conducted a sensitivity analysis of performance versus total simulation epochs. The results are shown in Tab.2 in the global rebuttal, tested on ScanNet++.
>
> The table shows that as the number of training epochs with physical loss increases, the stability of the objects improves. However, there is no significant improvement after 445 epochs, as most objects had already achieved stability. Notably, the longest training time was observed between 430 and 435 epochs, after which the time required for simulation progressively decreased, again due to the improved object stability.
>
> **5. The gradient backpropagated from SP-MC is fairly local and noisy. What if the initial shape is far from being stable?**
>
> While optimizing the implicit representation of the object shape, e.g., SDF, the surface points of the object are the most important. Therefore, we specifically sample surface points from the object  during optimization, following the common practice in related work (e.g., DeepSDF [1] Sec. 5 Data Preparation). In our method, the refinement stage of SP-MC accurately extract the surface points of the object, whose gradients are sufficient for optimizing the object's shape. Using all points from the coarse stage for backpropagation would result in excessive computations without significantly improving object shape optimization; the larger space outside the object surface are optimized with rendering loss in volume rendering.
>
> When the initial shape is far from being physically plausible, the shape will be improved locally but incrementally (shown in Fig.1 of global rebuttal), as the SP-MC sampling are conducted before every simulation. The physical loss ensures stability of the final result, together with our multi-stage training and reweighted rendering loss.

---

> ### Author Response · Authors · 2024-08-07
>
> **6. How is the ground-plane computed for physical simulation? Does the method handle complex terrian (e.g., with a carpet on the floor, or outdoor scenes)?**
>
> We use the supporting plane (i.e. the ground-plane in reviewer question) to provide contact and friction for object surface points in the physics simulation. For objects on the floor, we use SP-MC to obtain the surface points of both the object to be simulated and the background before each simulation. Additionally, we only include the background surface points located within proximity under the object surface points in the simulation to reduce computational load.
>
> This means our method is capable of handling more complex terrian, e.g., the carpet or outdoor scenes, if the corresponding SDFs are adequately trained.
>
> **7. The physical loss presented is not applicable to complex contact interactions, dynamic scenes or soft materials.**
>
> Yes, the current design of the physical loss and physical simulator in the paper is based on the assumption that rigid bodies move under the forces of gravity, contact, and friction. Thus, it cannot handle soft bodies (which deform after collision) or dynamic scenes. Extending the physical loss for broader applications and enhancing the physical simulator to support soft body collisions would be an interesting direction for future research. We will add this as an limitation and future potential in the revision.
>
> **8. Moving appendix C to the main text.**
>
> Due to the length limitation of the paper, we placed the implementation of the differentiable particle simulator in the appendix. Thank you for your suggestion, we will add relevant content to the main text in a future revision.
>
>
> [1] DeepSDF: Learning Continuous Signed Distance Functions for Shape Representation

---

> > ### Comment · Reviewer_fqR5 · 2024-08-13
> >
> > Thanks for the rebuttal, which addressed my questions. I'd like to keep my rating and recommend for accpetance.

---

### Official Review · Reviewer_mFxQ · 2024-07-10

**Soundness:** 3
**Presentation:** 4
**Contribution:** 3
**Rating:** 6
**Confidence:** 4

**Summary:**

This paper proposes to improve implicit surface representations by considering object dynamics (physics) in their resting state. It made a clever observation that the objects should be stable with respect to their supporting planes if the surface representation is correct. This observation leads to utilizing object surface points in the physical simulator to decide if the objects are stable, by penalizing any instability, the solution encourages more plausible surface reconstruction.

Physical loss and physical uncertainty were introduced, assisted by the proposed SP-MC (Surface Points Marching Cubes) to improve surface point sampling efficiency for deriving the losses in simulations.

Tests on ScanNet and ScanNet++ demonstrated superior results compared to existing works.

**Strengths:**

1. Applying physical stability loss is a rather novel idea.
2. The implementation of using particle-based physical simulation, although complicated, seems to be a plausible approach for understanding and observing any physical instability.
3. The paper is well-written with many implementation details.

**Weaknesses:**

1. Applying this idea would require an object mask, and also knowledge of the ground. This is not a constraint in the existing work MonoSDF.
2. The construction of Physical uncertainty (equation 7), based on the trajectory of the point before it makes contact, is not actually the same as the supporting structure of the object. To be more specific, such a trajectory is often a curve unless the object is falling without any rotation. Such a loss would lead to bulkiness in the supporting structure during reconstruction (as seen in Figure 7e).
3. The author mentioned a couple of times about the supporting plane or ground, where is that information from?
4. In general, this physical loss is only applicable to contact the the supporting plane, and not applicable to other contacts (such as a cup on a table).

**Questions:**

1. What is the ground plane, and how is it calculated/retrieved from the dataset?
2. In the losses the author used (as shown in the appendix), a combination of losses from many existing works is used, what would be the performance if the baseline is a version that combined all of the existing losses?
3. Section 3.3 mentioned the physical uncertainty $U_{phy}(r)$, which the author didn’t give a clear definition.

**Limitations:**

The author discussed certain limitations in the Appendix. See the above comments for additional concerns.

---

> ### Author Rebuttal · Authors · 2024-08-07
>
> We sincerely thank the reviewer for the constructive comments on our work. We will thoroughly address each of the concerns raised.
>
> **1. Our method requires object masks, whereas MonoSDF does not.**
>
> From neural scene reconstruction which treats the entire scene as a whole (e.g., MonoSDF[1]), adding object decompositionality is a popular trend as they can provide more valuable object-centric information for downstream applications like 3D reasoning, navigation or manipulation. Following prior work [2, 3, 4], we use object mask supervision to disentangle each object from the background. Separating objects in the scene further makes it possible to add physical simulations to each object. With the rapid development of segmentation and tracking models, e.g., SAM [5] and SAM2 [6], we can efficiently obtain object masks from images with off-the-shelf models, which could inspire more efforts in decompositional neural scene reconstruction.
>
> **2. The trajectory of contact point is not actually the same as the supporting structure of the object.**
>
> Updating physical uncertainty using contact trajectory may indeed deviate from the supporting structure. However, our proposed physical uncertainty is used to identify regions **potentially important for improving physical stability**, which is not used directly as a form of supervision. Our direct supervision comes from rendering supervision and the physical loss on the contact points only, with physical uncertainty playing only an auxiliary role in adjusting the loss weights and guiding pixel sampling towards these regions. As shown in our experiments and ablations, a coarse physical uncertainty is sufficient for improving slender-structure reconstruction and physical.
>
> **3. What is the supporting plane and how is it calculated?**
>
> We use the supporting plane to provide contact and friction for object surface points in the physics simulation. More specifically, we use SP-MC to obtain the surface points of both the object to be simulated and the supporting plane (e.g, background for objects on the floor) before each simulation. Additionally, we only include the background surface points located within proximity under the object surface points for the simulation to reduce computational load. We will add this in the revision.
>
> **4. Can physical loss be applied to more general contact scenarios, such as a cup on a table?**
>
> In fact, our physical simulator and physical loss both support more general contacts, as demonstrated in Fig. 1 and the supplementary video (e.g., the box on the cabinet and the monitor on the table). The only difference lies in the calculation method of the supporting plane during simulation. For example, for the monitor on the table, we use the table surface points instead of background surface points to calculate the supporting plane.
>
> **5. Add a baseline that combines all of the existing losses.**
>
> Our baseline (the first row, i.e., w/o RU, w/o PU, w/o PS, w/o PL) in Tab.2 combines all existing losses, which aligns with your case. Please refer to Tab.2 in the main paper for details.
>
> **6. Definition for physical uncertainty $U_{phy}(r)$.**
>
> Due to the length limitation of the paper, we briefly introduced $U_{phy}(r)$ in L194 as:
>
> "The physical uncertainty $U_{phy}(r)$ of the pixel corresponding to ray r is computed via volume rendering".
>
> Here we provide the following formula for calculating $U_{phy}(r)$:
>
> $$
> U_{phy}(r)=\sum_{i=0}^{n-1} T_{i}\alpha_{i}u_{phy,i}
> $$
>
>
> Thank you for your suggestion, we will add these definitions in the revision of the paper.
>
>
> [1] Monosdf: Exploring monocular geometric cues for neural implicit surface reconstruction
>
> [2] Objectsdf++: Improved object-compositional neural implicit surfaces
>
> [3] Rico: Regularizing the unobservable for indoor compositional reconstruction
>
> [4] Object-compositional neural implicit surfaces
>
> [5] Segment and Track Anything
>
> [6] SAM 2: Segment Anything in Images and Videos
>
> [7] Marching cubes: A high resolution 3d surface construction algorithm
>
> [8] A volumetric method for building complex models from range images

---

### Official Review · Reviewer_5HaF · 2024-07-13

**Soundness:** 3
**Presentation:** 3
**Contribution:** 2
**Rating:** 4
**Confidence:** 4

**Summary:**

This paper proposes a framework for scene reconstruction using neural implicit function with an emphasis on physical plausibility. The key idea is to integrate differentiable physical simulation into the neural scene reconstruction such that the reconstructed shape is stable under gravity. A surface points marching cube method is proposed to convert the neural surface into particles, which is differentiable, enabling the back propagation of gradients from simulation. The authors further propose rendering and physical uncertainty to identify the regions that are not consistent across multiview images and unstable under physical simulation. These uncertainties are beneficial to capture the thin structures of the reconstructed object. Experimental results on various datasets including ScanNet, ScanNet++,  and Replica are conducted to demonstrate the effectiveness of the proposed method.

**Strengths:**

- The paper is well-written. The specific techniques are thoroughly discussed with enough details provided. The proposed method is more like a system consisting of multiple components. The authors have clearly put a lot of efforts to elaborate on each component in detail, as can be seen in the appendix.

- The ablation study is nicely designed and well-controlled. I personally find Table 2 very informative, which clearly demonstrates the effectiveness of each proposed component.

**Weaknesses:**

- My major concern of the paper is its novelty. Though the authors position their paper as scene reconstruction, the examples shown in the paper are actually object-centric. In this sense, there are actually a series of works that have explored the physical stability of the generated objects represented as neural functions. Specifically, [37] handles the robustness of slender structure in shape generation, and [38] deals with stability of the generated shape. Especially the second work, it has a very similar problem setting to this paper, as they also first converted the SDF into surface points followed by a differentiable simulator. Somehow I feel the method in [38] is neater since the gradient is propagated to the shape latent code rather than a dense voxel grid as proposed in this paper. Neither [37] or [38] is compared in the experiments. This undermines the motivation and technical novelty of the proposed approach, since compared to shape generation, addressing scene reconstruction exclusively does not pose additional challenges for physical plausibility.

- With the above being said, I think the technical novelty lies in the uncertainty model. However, what confuses me is whether this modeling on the whole voxel grid is necessary. From the qualitative results in Fig. 7, the rendering uncertainty models are mainly helping to grow the shape regions that are "sparsely" covered by the multi-view images. I guess simple post-processing methods such as region growing would be sufficient for dealing with these cases. The physical uncertainty may also be unnecessary: from Fig. 7 and the failure cases in the appendix, physical uncertainty and physical losses are nothing more than changing the geometric shape so that its center of mass lies on the supporting plane of the shape's points connected to the ground. The proposed approach, especially the part of "physical uncertainty",  to me is an overkill to problem. I am also not a big fan of using the term "uncertainty" here since there is really no probability estimation/inference. A better wording could be "heuristics".

**Questions:**

See Weaknesses.

**Limitations:**

The authors have provided sufficient limitations and failure cases of the proposed method.

---

> ### Author Rebuttal · Authors · 2024-08-07
>
> We sincerely thank the reviewer for the constructive comments on our work. However, we believe the reviewer may have some misunderstandings regarding our method. We will clarify the novelty of our approach as the first effort to incorporate physical constraints into neural scene reconstruction, and we will further explain our rendering and physical uncertainty.
>
> **1. Misunderstandings about the method in our paper.**
>
> The reviewer frequently mentioned that we use a dense voxel grid, as noted in comments like "the gradient is propagated to a dense voxel grid as proposed in this paper" and "I think the technical novelty lies in the uncertainty model. However, what confuses me is whether this modeling on the whole voxel grid is necessary." In fact, our method uses MLPs to model SDF fields and rendering uncertainty simultaneously, while only physical uncertainty is modeled by a dense voxel grid as it reflects accurate 3D physical information from the simulator.
>
> Another misunderstanding comes from the rendering uncertainty, which addresses the **multi-view and multi-modal inconsistencies** in monocular geometry priors rather than "grow the shape regions that are 'sparsely' covered by the multi-view images". For more details, please refer to Section 3.3 in our paper.
>
> **2. The novelty of incorporating physical constraints into neural scene reconstruction based on the NeRF pipeline isn't limited.**
>
> We note that the reviewer considers our approach to be "object-centric" and confuses the difference between our work and [37, 38], which attempt to add physical constraints to object generation. This confusion extends to the novelty of our work, which integrates physical constraints into neural scene reconstruction. We will address these points one by one.
>
> First of all, our method is designed for decompositional scene reconstruction, which add object-centric concepts to neural scene reconstruction by simultaneous recovery of each object and the background. By disentangling objects in the scene, it offers object-centric information for holistic scene understanding and downstream applications like 3D reasoning, navigation or manipulation. In the meantime, it enables physical simulations for each object, which would not be possible for any simulation if we treated the entire scene as a whole.
>
> Second, we **argue there is a significant gap** between scene-level reconstruction (our work) and object-level generation ([37, 38]).
> * **Reconstruction v.s. Generation**: [37, 38] aim to generate physically plausible objects but do not replicate input scene images. Based on the DeepSDF[1] pipeline, [37, 38] do not use any image supervision or condition. In contrast, our work aims to recover a physically plausible 3D scene from given 2D images, ensuring the results match the original images.
> * **Scene-level v.s. Object-level**: Our approach leverages scene images, which are captured while walking freely within the scene, rather than taken around a specific object. The partial observations and abundant occlusions from the scene images make scene reconstruction much more challenging than object reconstruction.
>
> Therefore, it's impossible to directly adapt methods for object generation to scene reconstruction.
>
> Our work stands as the first to integrate differentiable rendering with differentiable physical simulations for learning implicit surface representations, thereby enhancing reconstruction quality and physical plausibility. This is achieved by our design of the SP-MC algorithm, an efficient differentiable particle-based simulator, an effective physical loss and joint uncertainty modeling, all of which contribute to the improved reconstruction quality and physical constraints stability. The technical novelty is acknowledged by other reviewers:
> * "Applying physical stability loss is a rather novel idea." - Reviewer mFxQ.
> * "There are good technical contributions and the method is sound." - Reviewer fqR5.
> * "The overall framework is mind-blowing and insightful." - Reviewer qV3n.
>
> **3. Rendering uncertainty isn't "mainly helping to grow the shape regions that are 'sparsely' covered by the multi-view images", and can't be substituted by "simple post-processing methods such as region growing".**
>
> Rendering uncertainty addresses inconsistencies in **monocular geometry priors** (depth and normal prior obtained from off-the-shelf geometry estimators) caused by **multi-view** (same region yielding different results in different views) and **multi-modal** (individual prediction of depth and normal cues leading to conflicts) issues. These inconsistencies arise from the limitations of geometry estimators and the prediction mechanism of monocular images.
>
> To the best of our knowledge, there are no post-processing methods that can efficiently identify these inconsistent regions. Plus, heuristic region growing may lead to degenerated results deviating from the image observations. On the contrary, our proposed rendering uncertainty, indirectly optimized by rendering loss during training, can locate the inconsistencies and mitigate them effectively.
>
> **4. Physical uncertainty isn't an overkill to problem.**
>
> We argue that our modeling of physical uncertainty is essential to reconstruct the slender structures and improve object stability. While physical loss can improve the stability of reconstruction, relying solely on it may achieve stability at the expense of the object's shape. The rendering uncertainty can mitigate monocular inconsistencies but they cannot identify regions crucial for physical stability. On the contrary, we design physical uncertainty to identify these regions by leveraging simulation and adjust rendering losses accordingly. Additionally, we use physical uncertainty-guided pixel sampling to enhance learning on these regions. These combined designs significantly improve the reconstruction results, especially in areas critical for object stability, as demonstrated in the ablations from Tab.2 main paper.

---

> > ### Comment · Reviewer_5HaF · 2024-08-13
> > **Reply to the rebuttal**
> >
> > Thank you for the authors’ rebuttal. Unfortunately, many of my initial concerns remain unaddressed. I would like to emphasize that the review and rebuttal process aims to improve the paper and make it compelling to a broader audience. If the paper is accepted, it should clearly convey the technical aspects of the proposed method: what is novel, what is comparable, and in which scenarios the method may fail. Therefore, I encourage the authors to consider my comments as potential questions that any general reader might ask. While these comments do not necessarily lead to the rejection of the paper, addressing them can significantly enhance the paper’s impact and solidity.
> >
> > Detailed responses to the authors’ rebuttal:
> >
> > **1. Use of MLPs for SDF fields and dense voxel grids**
> >
> > The authors state that their method employs MLPs to model SDF fields and rendering uncertainty, while a dense voxel grid is used solely for modeling physical uncertainty, reflecting accurate 3D physical information from the simulator. However, using MLPs for SDF does not inherently preclude the use of a dense voxel grid, as querying dense voxel positions is necessary for rendering images. My primary concern was whether the “dense voxel grid” is truly necessary, particularly when compared to the shape latent code used in [38]. I would be convinced if the authors provided results or compelling arguments demonstrating the advantages of the dense voxel grid over shape latent code, such as better reconstruction quality, faster convergence, or reduced computational resources.
> >
> > **2. Clarification on rendering uncertainty**
> >
> > The authors assert that my review implied that their method was “growing the shape regions that are sparsely covered by the multi-view images,” which was not true. My original concern was whether a simpler strategy, such as growing sparsely covered regions, might be sufficient to address the issue, rather than employing a complex model for “rendering uncertainty.”
> >
> > **3. Claim of integrating differentiable rendering with differentiable physical simulations**
> >
> > The authors claim that their work is the first to integrate differentiable rendering with differentiable physical simulations for learning implicit surface representation. I find this claim invalid, as there are already published works [1, 2] that achieve similar integration. Additionally, positive comments from other reviewers do not necessarily resolve the concerns I raised.
> >
> > [1] RISP: Rendering-Invariant State Predictor with Differentiable Simulation and Rendering for Cross-Domain Parameter Estimation, ICLR 2022.
> > [2] PAC-NeRF: Physics Augmented Continuum Neural Radiance Fields for Geometry-Agnostic System Identification, ICLR 2023.
> >
> > **4. Comparison between heuristic region growing and proposed method**
> >
> > The authors argue that heuristic region growing could lead to degenerated results deviating from the image observations. While this is true, it does not imply that the proposed method strictly adheres to image observations in all cases. A comparison between the two approaches would be necessary to substantiate this point.
> >
> > The remainder of the rebuttal has adequately addressed my concerns.

---

> ### Author Response · Authors · 2024-08-07
>
> **5. "Uncertainty" or "heuristics".**
>
> Although the uncertainty mentioned in this paper is not the same as the uncertainty in Bayesian learning, we use "uncertainty" in accordance with prior work [2, 3, 4] in NeRF and neural scene reconstruction.
>
> [1] DeepSDF: Learning Continuous Signed Distance Functions for Shape Representation
>
> [2] Activenerf: Learning where to see with uncertainty estimation
>
> [3] Conditional-flow nerf: Accurate 3d modelling with reliable uncertainty quantification
>
> [4] Debsdf: Delving into the details and bias of neural indoor scene reconstruction
>
> [37] Physical simulation layer for accurate 3d modeling
>
> [38] Physically-aware generative network for 3d shape modeling

---

> ### Author Response · Authors · 2024-08-14
> **More discussion**
>
> Thank you for the suggestions. We have provided further clarification below and will incorporate the discussions in the revision.
>
> **1. Use of MLPs for SDF fields and dense voxel grids**
>
> Thanks for further clarifying your primary concern, which suggests that the shape latent code could be used to further achieve physically plausible object reconstruction for each individual object, similarly to the auto-decoder in [38]. However, we believe that this method may present the following three issues:
> * The optimization for the shape code in [38] does not allow for the incorporation of image conditions, and directly integrating rendering loss remains unexplored. In contrast, our method can seamlessly incorporate the rendering losses.
> * The above method using shape latent codes requires querying two representations for rendering images or exporting a mesh. In contrast, our method only learns a single representation through NeRF MLPs, which is the sole representation queried during inference, making it more streamlined. The dense voxel grid is only used during training to adjust the loss and guide pixel sampling, without affecting SDF calculation.
> * The 3D geometry obtained using the shape latent code is constrained by the generalizability of the pre-trained auto-decoder, which converts the code into an object's SDF. Our method, on the other hand, can effectively overfit the scene without the need for extra data priors.
>
> Thanks again for your comments, we will add more clarification in the revision.
>
> **2. Clarification on rendering uncertainty**
>
> Thanks for further clarifying your original concern. We believe that the simple strategy, e.g., 'growing sparsely covered regions', may struggle to effectively address the issues in reconstructing the thin structures, primarily due to two key challenges:
>
> * **How to locate the less reconstructed regions (e.g., sparsely covered)?** Generally it's non-trivial to identify the 3D regions that require further attention. For example, determining the 3D points visible from images is error-pruning (considering the error in the up-to-scale depth estimation and camera poses),  computationally intensive (enumeration over the 3D space for all input views), and there is no intuitive method to quantify their sparsity. On the other hand, the rendering uncertainties have been proposed to implicitly achieve this goal from the rendering losses perspective. The physical uncertainty also provides explicit and efficient knowledge of where potentially lacks structure from the external knowledge in the simulation.
> * **How to grow these sparsely covered regions?** Designing a heuristic region-growing mechanism within the context of implicit representation can also be challenging, as it may easily lead to degenerated results. This is also the reason why we rely on modulating the rendering loesses to recover the 3D geometry, and only apply physical losses on the contact points. How to effectively grow the region under the image constraints may be a separate problem by itself.
>
> In summary, we believe that the solution to effectively leverage 'simple strategy' is not as intuitive as it may look, and its extension may require extra design choices and hyper-parameter search, which are out of the scope of this paper. However, we will be more than happy to add this discussion to the revision if you have other effective strategies to solve this problem in mind.
>
> **3. Claim of integrating differentiable rendering with differentiable physical simulations**
>
> We clarify our method is distinguished by optimizing the geometry shape using both differentiable rendering and differentiable physical simulations. In contrast, RISP[1] does not optimize object geometry at all, instead, it focuses on controlling given objects within a differentiable physical environment and integrates differentiable rendering into the simulation to improve the generalizability of object control in unknown rendering configurations. PAC-NeRF[2], like vanilla NeRF[3], uses only differentiable rendering to optimize object geometry shape. They employ a physical simulator to optimize the physical properties (such as elastic materials, plasticine, sand, and Newtonian/non-Newtonian fluids) of objects rather than the geometry.
>
> While there are existing works that leverage both differentiable rendering and simulation, our submission and rebuttal describe our work as "the first method to integrate differentiable rendering with differentiable physical simulation *for neural surface reconstruction*." We will add these discussions in the related work section in the revision.

---

> ### Author Response · Authors · 2024-08-14
> **More discussion (Cont.)**
>
> **4. Comparison between heuristic region growing and proposed method**
>
> While the goal of reconstruction is to recover the 3D geometry according to the image inputs, we believe existing methods are far from "strictly adhering to image observations in all cases," especially for the less observed ones. In our work, we have made efforts to incorporate uncertainties into the rendering losses to mitigate this gap. On the other hand, how to perform "heuristic region growing" is not trivial (see our discussion in Question 2), and its integration with rendering losses remains unexplored. We consider this comparison to be beyond the scope of the current paper, but we will include this discussion in the revision if additional suggestions arise.
>
>
> [1] RISP: Rendering-Invariant State Predictor with Differentiable Simulation and Rendering for Cross-Domain Parameter Estimation.
> [2] PAC-NeRF: Physics Augmented Continuum Neural Radiance Fields for Geometry-Agnostic System Identification.
> [3] NeRF: Representing Scenes as Neural Radiance Fields for View Synthesis.
> [38] Physically-aware generative network for 3d shape modeling.

---

### Author Rebuttal · Authors · 2024-08-07

We sincerely thank all the reviewers for their efforts in reviewing this manuscript and for their constructive comments. We particularly appreciate their recognition of our proposed framework as "mind-blowing, insightful" (qV3n) and "novel" (fqR5, mFxQ, qV3N), with "extensive experiments and ablations" (5HaF, qV3n, fqR5) demonsrating "strong results" (fqR5) and "significant improvements" (qV3n). Additionally, the reviewers recognize our paper as "well written and easy to follow with enough implementation details" (5HaF, mFxQ, fqR5).

Below, we address the questions from the reviewers respectively. Per the reviewers' requests, we also provide a additional PDF page in the global rebuttal that includes:
* Quantitative comparisons of varying maximum simulation steps in a forward simulation.
* Quantitative results from increasing the number of training epochs with physical simulations.
* Visualization of intermediate results during training.

We hope these clarifications enhance the reviewers' confidence in evaluating our work. We will integrate all the feedbacks to complement the original draft. Together, we are confident in delivering a high-quality paper beneficial to the research community.

---

### Decision · Program_Chairs · 2024-09-25

**Decision:**

Accept (poster)

**Comment:**

The paper presents PhyRecon, a method combining differentiable rendering and physics simulation to improve 3D reconstruction using neural implicit representations. It introduces Surface Points Marching Cubes (SP-MC) to enhance surface detail and accounts for uncertainties to improve realism and stability. This approach integrates physics simulation with rendering for better 3D reconstruction and significantly improves reconstruction quality and stability. The reviews have concerns on the approach might be too complex, and focuses only on static, rigid objects. After carefully reading the paper, reviews, and rebuttals, the AC recommends to accept the paper.